# GROUP EQUIVARIANT
# STAND-ALONE SELF-ATTENTION FOR VISION

**David W. Romero**
Vrije Universiteit Amsterdam
d.w.romeroguzman@vu.nl

**Jean-Baptiste Cordonnier**
École Polytechnique Fédérale de Lausanne (EPFL)
jean-baptiste.cordonnier@epfl.ch

## ABSTRACT

We provide a general self-attention formulation to impose group equivariance to arbitrary symmetry groups. This is achieved by defining positional encodings that are invariant to the action of the group considered. Since the group acts on the positional encoding directly, group equivariant self-attention networks (`GSA-Nets`) are steerable by nature. Our experiments on vision benchmarks demonstrate consistent improvements of `GSA-Nets` over non-equivariant self-attention networks.

## 1 INTRODUCTION

Recent advances in Natural Language Processing have been largely attributed to the rise of the *Transformer* (Vaswani et al., 2017). Its key difference with previous methods, e.g., recurrent neural networks, convolutional neural networks (CNNs), is its ability to query information from all the input words simultaneously. This is achieved via the *self-attention operation* (Bahdanau et al., 2015; Cheng et al., 2016), which computes the similarity between representations of words in the sequence in the form of *attention scores*. Next, the representation of each word is updated based on the words with the highest attention scores. Inspired by the capacity of transformers to learn meaningful inter-word dependencies, researchers have started applying self-attention in vision tasks. It was first adopted into CNNs by channel-wise attention (Hu et al., 2018) and non-local spatial modeling (Wang et al., 2018). More recently, it has been proposed to replace CNNs with self-attention networks either partially (Bello et al., 2019) or entirely (Ramachandran et al., 2019). Contrary to discrete convolutional kernels, weights in self-attention are not tied to particular positions (Fig. A.1), yet self-attention layers are able to express any convolutional layer (Cordonnier et al., 2020). This flexibility allows leveraging long-range dependencies under a fixed parameter budget.

An arguable orthogonal advancement to deep learning architectures is the incorporation of symmetries into the model itself. The seminal work by Cohen & Welling (2016) provides a recipe to extend the *translation equivariance* of CNNs to other symmetry groups to improve generalization and sample-efficiency further (see §2). *Translation equivariance* is key to the success of CNNs. It describes the property that if a pattern is translated, its numerical descriptors are also translated, but not modified.

In this work, we introduce *group self-attention*, a self-attention formulation that grants equivariance to arbitrary symmetry groups. This is achieved by defining positional encodings invariant to the action of the group considered. In addition to generalization and sample-efficiency improvements provided by group equivariance, group equivariant self-attention networks (`GSA-Nets`) bring important benefits over group convolutional architectures: (*i*) *Parameter efficiency:* contrary to conventional discrete group convolutional kernels, where weights are tied to particular positions of neighborhoods on the group, group equivariant self-attention leverages long-range dependencies on group functions under a fixed parameter budget, yet it is able to express any group convolutional kernel. This allows for very expressive networks with low parameter count. (*ii*) *Steerability:* since the group acts directly on the positional encoding, `GSA-Nets` are *steerable* (Weiler et al., 2018b) by nature. This allows us to go beyond group discretizations that live in the grid without introducing interpolation artifacts.

**Contributions:**

- We provide an extensive analysis on the equivariance properties of self-attention (§4).
- We provide a general formulation to impose group equivariance to self-attention (§5).
- We provide instances of self-attentive architectures equivariant to several symmetry groups (§6).
- Our results demonstrate consistent improvements of `GSA-Nets` over non-equivariant ones (§6).

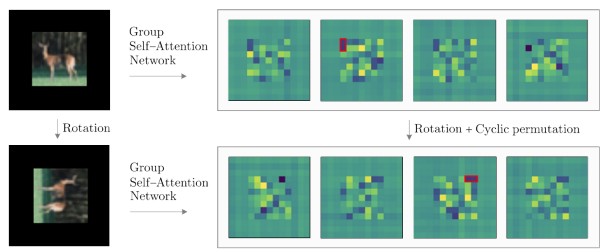

Figure 1: Behavior of feature representations in group self-attention networks. An input rotation induces a rotation plus a cyclic permutation to the intermediary feature representations of the network. Additional examples for all the groups used in this work as well as their usage are provided in `repo/demo/`.

## 2 RELATED WORK

Several approaches exist which provide equivariance to various symmetry groups. The translation equivariance of CNNs has been extended to additional symmetries ranging from planar rotations (Dieleman et al., 2016; Marcos et al., 2017; Worrall et al., 2017; Weiler et al., 2018b; Li et al., 2018; Cheng et al., 2018; Hoogeboom et al., 2018; Bekkers et al., 2018; Veeling et al., 2018; Lenssen et al., 2018; Graham et al., 2020) to spherical rotations (Cohen et al., 2018; 2019b; Worrall & Brostow, 2018; Weiler et al., 2018a; Esteves et al., 2019a;b; 2020), scaling (Marcos et al., 2018; Worrall & Welling, 2019; Sosnovik et al., 2020; Romero et al., 2020b) and more general symmetry groups (Cohen & Welling, 2016; Kondor & Trivedi, 2018; Tai et al., 2019; Weiler & Cesa, 2019; Cohen et al., 2019a; Bekkers, 2020; Venkataraman et al., 2020). Importantly, all these approaches utilize discrete convolutional kernels, and thus, tie weights to particular positions in the neighborhood on which the kernels are defined. As group neighborhoods are (much) larger than conventional ones, the number of weights discrete group convolutional kernels require proportionally increases. This phenomenon is further exacerbated by *attentive group equivariant networks* (Romero & Hoogendoorn, 2019; Diaconu & Worrall, 2019; Romero et al., 2020a). Since attention is used to leverage non-local information to aid local operations, non-local neighborhoods are required. However, as attention branches often rely on discrete convolutions, they effectively tie specific weights to particular positions on a large non-local neighborhood on the group. As a result, attention is bound to growth of the model size, and thus, to negative statistical efficiency. Differently, group self-attention is able to attend over arbitrarily large group neighborhoods under a fixed parameter budget. In addition, group self-attention is steerable by nature (§5.1) a property primarily exhibited by works carefully designed to that end.

Other way to detach weights from particular positions comes by parameterizing convolutional kernels as (constrained) neural networks (Thomas et al., 2018; Finzi et al., 2020). Introduced to handle irregularly-sampled data, e.g., point-clouds, networks parameterizing convolutional kernels receive relative positions as input and output their values at those positions. In contrast, our mappings change as a function of the input content. Most relevant to our work are the *SE(3)* and *Lie* Transformers (Fuchs et al., 2020; Hutchinson et al., 2020). However, we obtain group equivariance via a generalization of positional encodings, Hutchinson et al. (2020) does so via operations on the Lie algebra, and Fuchs et al. (2020) via irreducible representations. In addition, our work prioritizes applications on visual data and extensively analyses theoretical aspects and properties of group equivariant self-attention.

## 3 STAND-ALONE SELF-ATTENTION

In this section, we recall the mathematical formulation of self-attention and emphasize the role of the positional encoding. Next, we introduce a functional formulation to self-attention which will allow us to analyze and generalize its equivariance properties.

**Definition.** Let $\mathbf{X} \in \mathbb{R}^{N \times C_{\text{in}}}$ be an input matrix consisting of $N$ tokens of $C_{\text{in}}$ dimensions each.[1] A self-attention layer maps an input matrix $\mathbf{X} \in \mathbb{R}^{N \times C_{\text{in}}}$ to an output matrix $\mathbf{Y} \in \mathbb{R}^{N \times C_{\text{out}}}$ as:

$$\mathbf{Y} = \text{SA}(\mathbf{X}) \coloneqq \text{softmax}_{[\,,:]}(\mathbf{A})\mathbf{X}\mathbf{W}_{\text{val}}, \tag{1}$$

with $\mathbf{W}_{\text{val}} \in \mathbb{R}^{C_{\text{in}} \times C_{\text{h}}}$ the *value matrix*, $\mathbf{A} \in \mathbb{R}^{N \times N}$ the *attention scores matrix*, and $\text{softmax}_{[\,,:]}(\mathbf{A})$ the *attention probabilities*. The matrix $\mathbf{A}$ is computed as:

$$\mathbf{A} \coloneqq \mathbf{X}\mathbf{W}_{\text{qry}}(\mathbf{X}\mathbf{W}_{\text{key}})^{\top}, \tag{2}$$

parameterized by *query* and *key matrices* $\mathbf{W}_{\text{qry}}, \mathbf{W}_{\text{key}} \in \mathbb{R}^{C_{\text{in}} \times C_{\text{h}}}$. In practice, it has been found beneficial to apply multiple self-attention operations, also called *heads*, in parallel, such that different

---

[1]We consequently consider an image as a set of $N$ discrete objects $i \in \{1, 2, ..., N\}$.

heads are able to attend to different parts of the input. In this *multi-head self-attention* formulation, the output of $H$ heads of output dimension $C_h$ are concatenated and projected to $C_{\text{out}}$ as:

$$\text{MHSA}(\mathbf{X}) \coloneqq \underset{h \in [H]}{\text{concat}} \big[ \text{SA}^{(h)}(\mathbf{X}) \big] \mathbf{W}_{\text{out}} + \mathbf{b}_{\text{out}}, \tag{3}$$

with a *projection matrix* $\mathbf{W}_{\text{out}} \in \mathbb{R}^{HC_h \times C_{\text{out}}}$ and a bias term $\mathbf{b}_{\text{out}} \in \mathbb{R}^{C_{\text{out}}}$.

## 3.1 THE ROLE OF THE POSITIONAL ENCODING

Note that the self-attention operation defined in Eq. 3 is equivariant to permutations of the input rows of $\mathbf{X}$. That is, a permutation of the rows of $\mathbf{X}$ will produce the same output $\mathbf{Y}$ up to this permutation. Hence, self-attention is blind to the order of its inputs, i.e., it is a set operation. Illustratively, an input image is processed as a bag of pixels and the structural content is not considered. To alleviate this limitation, the input representations in self-attention are often enriched with a *positional encoding* that provides positional information of the set elements.

**Absolute positional encoding.** Vaswani et al. (2017) introduced a (learnable) positional encoding $\mathbf{P} \in \mathbb{R}^{N \times C_{\text{in}}}$ for each input position which is added to the inputs when computing the attention scores:

$$\mathbf{A} \coloneqq (\mathbf{X} + \mathbf{P}) \mathbf{W}_{\text{qry}} ((\mathbf{X} + \mathbf{P}) \mathbf{W}_{\text{key}})^{\top}. \tag{4}$$

More generally, $\mathbf{P}$ can be substituted by any function that returns a vector representation of the position and can be incorporated by means of addition or concatenation, e.g., Zhao et al. (2020). This positional encoding injects additional structural information about the tokens into the model, which makes it susceptible to changes in the token's positions. Unfortunately, the model must learn to recognize similar patterns at every position independently as absolute positional encodings are *unique* to each position. This undesired data inefficiency is addressed by *relative positional encodings*.

**Relative positional encoding.** Introduced by Shaw et al. (2018), relative encodings consider the *relative distance* between the query token $i$ – the token we compute the representation of –, and the key token $j$ – the token we attend to –. The calculation of the attention scores (Eq. 2) then becomes:

$$\mathbf{A}_{i,j}^{\text{rel}} \coloneqq \mathbf{X}_i \mathbf{W}_{\text{qry}} ((\mathbf{X}_j + \mathbf{P}_{x(j)-x(i)}) \mathbf{W}_{\text{key}})^{\top}, \tag{5}$$

where $\mathbf{P}_{x(j)-x(i)} \in \mathbb{R}^{1 \times C_{\text{in}}}$ is a vector representation of the relative shift and $x(i)$ is the position of the token $i$ as defined in §3.2. Consequently, similar patterns can be recognized at arbitrary positions, as relative query-key distances always remain equal.

## 3.2 A FUNCTIONAL FORMULATION TO SELF-ATTENTION

**Notation.** We denote by $[n]$ the set $\{1, 2, \ldots, n\}$. Given a set $\mathcal{S}$ and a vector space $\mathcal{V}$, $L_V(\mathcal{S})$ will denote the space of functions $\{f : \mathcal{S} \to \mathcal{V}\}$. Square brackets are used when functions are arguments.

Let $\mathcal{S} = \{i\}_{i=1}^{N}$ be a set of $N$ elements. A matrix $\mathbf{X} \in \mathbb{R}^{N \times C_{\text{in}}}$ can be interpreted as a vector-valued function $f : \mathcal{S} \to \mathbb{R}^{C_{\text{in}}}$ that maps element sets $i \in \mathcal{S}$ to $C_{\text{in}}$-dimensional vectors: $f : i \mapsto f(i)$. Consequently, a matrix multiplication, $\mathbf{X} \mathbf{W}_y^{\top}$, of matrices $\mathbf{X} \in \mathbb{R}^{N \times C_{\text{in}}}$ and $\mathbf{W}_y \in \mathbb{R}^{C_{\text{out}} \times C_{\text{in}}}$ can be represented as a function $\varphi_y : L_{V_{C_{\text{in}}}}(\mathcal{S}) \to L_{V_{C_{\text{out}}}}(\mathcal{S})$, $\varphi_y : f(i) \mapsto \varphi_y(f(i))$, parameterized by $\mathbf{W}_y$, between functional spaces $L_{V_{C_{\text{in}}}}(\mathcal{S}) = \{f : \mathcal{S} \to \mathbb{R}^{C_{\text{in}}}\}$ and $L_{V_{C_{\text{out}}}}(\mathcal{S}) = \{f : \mathcal{S} \to \mathbb{R}^{C_{\text{out}}}\}$. Following this notation, we can represent the position-less attention scores calculation (Eq. 2) as:

$$\mathbf{A}_{i,j} = \alpha[f](i,j) = \langle \varphi_{\text{qry}}(f(i)), \varphi_{\text{key}}(f(j)) \rangle. \tag{6}$$

The function $\alpha[f] : \mathcal{S} \times \mathcal{S} \to \mathbb{R}$ maps pairs of set elements $i, j \in \mathcal{S}$ to the *attention score* of $j$ relative to $i$. Therefore, the self-attention (Eq. 1) can be written as:

$$\mathbf{Y}_{i,:} = \zeta[f](i) = \sum_{j \in \mathcal{S}} \sigma_j(\alpha[f](i,j)) \varphi_{\text{val}}(f(j)) = \sum_{j \in \mathcal{S}} \sigma_j(\langle \varphi_{\text{qry}}(f(i)), \varphi_{\text{key}}(f(j)) \rangle) \varphi_{\text{val}}(f(j)), \tag{7}$$

where $\sigma_j = \text{softmax}_j$ and $\zeta[f] : \mathcal{S} \to \mathbb{R}^{C_h}$. Finally, multi-head self-attention (Eq. 3) can be written as:

$$\text{MHSA}(\mathbf{X})_{i,:} = m[f](i) = \varphi_{\text{out}}\Big( \bigcup_{h \in [H]} \zeta^{(h)}[f](i) \Big)$$

$$= \varphi_{\text{out}}\Big( \bigcup_{h \in [H]} \sum_{j \in \mathcal{S}} \sigma_j(\langle \varphi_{\text{qry}}^{(h)}(f(i)), \varphi_{\text{key}}^{(h)}(f(j)) \rangle) \varphi_{\text{val}}^{(h)}(f(j)) \Big), \tag{8}$$

where $\cup$ is the functional equivalent of the concatenation operator $\text{concat}$, and $m[f] : \mathcal{S} \to \mathbb{R}^{C_{\text{out}}}$.

**Local self-attention.** Recall that $\alpha[f]$ assigns an attention scores to every other set element $j \in \mathcal{S}$ relative to the query element $i$. The computational cost of self-attention is often reduced by restricting

its calculation to a local neighborhood $\mathcal{N}(i)$ around the query token $i$ analogous in nature to the local receptive field of CNNs (Fig. A.1a). Consequently, *local self-attention* can be written as:

$$m[f](i) = \varphi_{\text{out}}\Big( \bigcup_{h \in [H]} \sum_{j \in \mathcal{N}(i)} \sigma_j\big(\langle \varphi_{\text{qry}}^{(h)}(f(i)), \varphi_{\text{key}}^{(h)}(f(j)) \rangle\big) \varphi_{\text{val}}^{(h)}(f(j)) \Big). \tag{9}$$

Note that Eq. 9 is equivalent to Eq. 8 for $\mathcal{N}(i) = \mathcal{S}$, i.e. when considering global neighborhoods.

**Absolute positional encoding.** The absolute positional encoding is a function $\rho : \mathcal{S} \to \mathbb{R}^{C_{\text{in}}}$ that maps set elements $i \in \mathcal{S}$ to a vector representation of its position: $\rho : i \to \rho(i)$. Note that this encoding is not dependent on functions defined on the set but *only* on the set itself.[2] Consequently, absolute position-aware self-attention (Eq. 4) can be written as:

$$m[f, \rho](i) = \varphi_{\text{out}}\Big( \bigcup_{h \in [H]} \sum_{j \in \mathcal{N}(i)} \sigma_j\big(\langle \varphi_{\text{qry}}^{(h)}(f(i) + \rho(i)), \varphi_{\text{key}}^{(h)}(f(j) + \rho(j)) \rangle\big) \varphi_{\text{val}}^{(h)}(f(j)) \Big). \tag{10}$$

The function $\rho$ can be decomposed as two functions $\rho^P \circ x$: (*i*) the *position function* $x : \mathcal{S} \to \mathcal{X}$, which provides the position of set elements in the underlying space $\mathcal{X}$ (e.g., pixel positions), and, (*ii*) the *positional encoding* $\rho^P : \mathcal{X} \to \mathbb{R}^{C_{\text{in}}}$, which provides vector representations of elements in $\mathcal{X}$. This distinction will be of utmost importance when we pinpoint where exactly (group) equivariance must be imposed to the self-attention operation (§4.3, §5).

**Relative positional encoding.** Here, positional information is provided in a relative manner. That is, we now provide vector representations of relative positions $\rho(i, j) := \rho^P(x(j) - x(i))$ among pairs $(i, j)$, $i \in \mathcal{S}, j \in \mathcal{N}(i)$. Consequently, relative position-aware self-attention (Eq. 5) can be written as:

$$m^r[f, \rho](i) = \varphi_{\text{out}}\Big( \bigcup_{h \in [H]} \sum_{j \in \mathcal{N}(i)} \sigma_j\big(\langle \varphi_{\text{qry}}^{(h)}(f(i)), \varphi_{\text{key}}^{(h)}(f(j) + \rho(i, j)) \rangle\big) \varphi_{\text{val}}^{(h)}(f(j)) \Big). \tag{11}$$

## 4 EQUIVARIANCE ANALYSIS OF SELF-ATTENTION

In this section we analyze the equivariance properties of self-attention. Since the analysis largely relies on group theory, we provide all concepts required for proper understanding in Appx. C.

### 4.1 GROUP EQUIVARIANCE AND EQUIVARIANCE FOR FUNCTIONS DEFINED ON SETS

First we provide the general definition of group equivariance and refine it to relevant groups next. Additionally, we define the property of *unique equivariance* to restrict equivariance to a given group.

**Definition 4.1** (**Group equivariance**). *Let $\mathcal{G}$ be a group (Def. C.1), $\mathcal{S}, \mathcal{S}'$ be sets, $\mathcal{V}, \mathcal{V}'$ be vector spaces, and $\mathcal{L}_g[\cdot], \mathcal{L}'_g[\cdot]$ be the induced (left regular) representation (Def. C.4) of $\mathcal{G}$ on $L_\mathcal{V}(\mathcal{S})$ and $L_{\mathcal{V}'}(\mathcal{S}')$, respectively. We say that a map $\varphi : L_\mathcal{V}(\mathcal{S}) \to L_{\mathcal{V}'}(\mathcal{S}')$ is equivariant to the action of $\mathcal{G}$ – or $\mathcal{G}$-equivariant –, if it commutes with the action of $\mathcal{G}$. That is, if:*

$$\varphi\big[\mathcal{L}_g[f]\big] = \mathcal{L}'_g\big[\varphi[f]\big], \quad \forall f \in L_\mathcal{V}(\mathcal{S}), \ \forall g \in \mathcal{G}.$$

**Example 4.1.1** (**Permutation equivariance**). *Let $\mathcal{S} = \mathcal{S}' = \{i\}_{i=1}^N$ be a set of $N$ elements, and $\mathcal{G} = \mathbb{S}_N$ be the group of permutations on sets of $N$ elements. A map $\varphi : L_\mathcal{V}(\mathcal{S}) \to L_{\mathcal{V}'}(\mathcal{S})$ is said to be equivariant to the action of $\mathbb{S}_N$ – or permutation equivariant –, if:*

$$\varphi\big[\mathcal{L}_\pi[f]\big](i) = \mathcal{L}'_\pi\big[\varphi[f]\big](i), \quad \forall f \in L_\mathcal{V}(\mathcal{S}), \ \forall \pi \in \mathbb{S}_N, \ \forall i \in \mathcal{S},$$

*where $\mathcal{L}_\pi[f](i) := f(\pi^{-1}(i))$, and $\pi : \mathcal{S} \to \mathcal{S}$ is a bijection from the set to itself. The element $\pi(i)$ indicates the index to which the $i$-th element of the set is moved to as an effect of the permutation $\pi$. In other words, $\varphi$ is said to be permutation equivariant if it commutes with permutations $\pi \in \mathbb{S}_N$. That is, if permutations in its argument produce equivalent permutations on its response.*

Several of the transformations of interest, e.g., rotations, translations, are not defined on sets. Luckily, as we consider sets gathered from homogeneous spaces $\mathcal{X}$ where these transformations are well-defined, e.g., $\mathbb{R}^2$ for pixels, there exists an injective map $x : \mathcal{S} \to \mathcal{X}$ that associates a position in $\mathcal{X}$ to each set element, the *position function*. In Appx. D we show that the action of $\mathcal{G}$ on such a set is well-defined and induces a group representation to functions on it. With this in place, we are now able to define equivariance of set functions to groups whose actions are defined on homogeneous spaces.

**Definition 4.2** (**Equivariance of set functions to groups acting on homogeneous spaces**). *Let $\mathcal{G}$ be a group acting on two homogeneous spaces $\mathcal{X}$ and $\mathcal{X}'$, let $\mathcal{S}, \mathcal{S}'$ be sets and $\mathcal{V}, \mathcal{V}'$ be vector*

---

[2]Illustratively, one can think of this as a function returning a vector representation of pixel positions in a grid. Regardless of any transformation performed to the image, the labeling of the grid itself remains exactly equal.

spaces. Let $x : \mathcal{S} \to \mathcal{X}$ and $x' : \mathcal{S}' \to \mathcal{X}'$ be injective maps. We say that a map $\varphi : L_V(\mathcal{S}) \to L_{V'}(\mathcal{S}')$ is equivariant to the action of $\mathcal{G}$ – or $\mathcal{G}$-equivariant –, if it commutes with the action of $\mathcal{G}$. That is, if:

$$\varphi\big[\mathcal{L}_g[f]\big] = \mathcal{L}'_g\big[\varphi[f]\big], \quad \forall f \in L_V(\mathcal{S}), \ \forall g \in \mathcal{G},$$

where $\mathcal{L}_g[f](i) \coloneqq f(x^{-1}(g^{-1}x(i)))$, $\mathcal{L}'_g[f](i) \coloneqq f(x'^{-1}(g^{-1}x'(i)))$ are the induced (left regular) representation of $\mathcal{G}$ on $L_V(\mathcal{S})$ and $L_{V'}(\mathcal{S}')$, respectively. I.o.w., $\varphi$ is said to be $\mathcal{G}$-equivariant if a transformation $g \in \mathcal{G}$ on its argument produces a corresponding transformation on its response.

**Example 4.2.1** (**Translation equivariance**). *Let $\mathcal{S}, \mathcal{S}'$ be sets and let $x : \mathcal{S} \to \mathcal{X}$ and $x' : \mathcal{S}' \to \mathcal{X}'$ be injective maps from the sets $\mathcal{S}, \mathcal{S}'$ to the corresponding homogeneous spaces $\mathcal{X}, \mathcal{X}'$ on which they are defined, e.g., $\mathbb{R}^d$ and $\mathcal{G}$. With $(\mathcal{X}, +)$ the translation group acting on $\mathcal{X}$, we say that a map $\varphi : L_V(\mathcal{S}) \to L_{V'}(\mathcal{S})$ is equivariant to the action of $(\mathcal{X}, +)$ – or translation equivariant –, if:*

$$\varphi\big[\mathcal{L}_y[f]\big](i) = \mathcal{L}'_y\big[\varphi[f]\big](i), \quad \forall f \in L_V(\mathcal{S}), \ \forall y \in \mathcal{X},$$

*with $\mathcal{L}_y[f](i) \coloneqq f(x^{-1}(x(i)-y))$, $\mathcal{L}'_y[f](i) \coloneqq f(x'^{-1}(x'(i)-y))$. I.o.w., $\varphi$ is said to be translation equivariant if a translation on its argument produces a corresponding translation on its response.*

## 4.2 Equivariance properties of self-attention

In this section we analyze the equivariance properties of the self-attention. The proofs to all the propositions stated in the main text are provided in Appx. G.

**Proposition 4.1.** *The global self-attention formulation without positional encoding (Eqs. 3, 8) is permutation equivariant. That is, it holds that: $m[\mathcal{L}_\pi[f]](i) = \mathcal{L}_\pi[m[f]](i)$.*

Note that permutation equivariance only holds for global self-attention. The local variant proposed in Eq. 9 reduces permutation equivariance to a smaller set of permutations where neighborhoods are conserved under permutation, i.e., $\mathbb{S}_n = \{\pi \in \mathbb{S}_N \mid j \in \mathcal{N}(i) \to \pi(j) \in \mathcal{N}(i), \ \forall i \in \mathcal{S}\}$.

**Permutation equivariance induces equivariance to important (sub)groups.** Consider the cyclic group of order 4, $\mathbb{Z}_4 = \{e, r, r^2, r^3\}$ which induces planar rotations by $90°$.[3] As every rotation in $\mathbb{Z}_4$ effectively induces a permutation of the tokens positions, it can be shown that $\mathbb{Z}_4$ is a subgroup of $\mathbb{S}_N$, i.e., $\mathbb{S}_N \geq \mathbb{Z}_4$. Consequently, maps equivariant to permutations are automatically equivariant to $\mathbb{Z}_4$. However, as the permutation equivariance constraint is harder than that of $\mathbb{Z}_4$-equivariance, imposing $\mathbb{Z}_4$-equivariance as a result of permutation equivariance is undesirable in terms of expressivity. Consequently, Ravanbakhsh et al. (2017) introduced the concept of *unique $\mathcal{G}$-equivariance* to express the family of functions equivariant to $\mathcal{G}$ but not equivariant to other groups $\mathcal{G}' \geq \mathcal{G}$:

**Definition 4.3** (**Unique $\mathcal{G}$-equivariance**). *Let $\mathcal{G}$ a subgroup of $\mathcal{G}'$, $\mathcal{G} \leq \mathcal{G}'$ (Def. C.2). We say that a map $\varphi$ is uniquely $\mathcal{G}$-equivariant iff it is $\mathcal{G}$-equivariant but not $\mathcal{G}'$-equivariant for any $\mathcal{G}' \geq \mathcal{G}$.*

In the following sections, we show that we can enforce unique equivariance not only to subgroups of $\mathbb{S}_N$, e.g., $\mathbb{Z}_4$, but also to other interesting groups not contained in $\mathbb{S}_N$, e.g., groups of rotations finer than 90 degrees. This is achieved by enriching set functions with a proper positional encoding.

**Proposition 4.2.** *Absolute position-aware self-attention (Eqs. 4, 10) is neither permutation nor translation equivariant. i.e., $m[\mathcal{L}_\pi[f], \rho](i) \neq \mathcal{L}_\pi[m[f, \rho]](i)$ and $m[\mathcal{L}_y[f], \rho](i) \neq \mathcal{L}_y[m[f, \rho]](i)$.*

Though absolute positional encodings do disrupt permutations equivariance, they are unable to provide translation equivariance. We show next that translation equivariance is obtained via relative encodings.

**Proposition 4.3.** *Relative position-aware self-attention (Eq. 11) is translation equivariant. That is, it holds that: $m^r[\mathcal{L}_y[f], \rho](i) = \mathcal{L}_y[m^r[f, \rho]](i)$.*

## 4.3 Where exactly is equivariance imposed in self-attention?

In the previous section we have seen two examples of successfully imposing group equivariance to self-attention. Specifically, we see that no positional encoding allows for permutation equivariance and that a relative positional encoding allows for translation equivariance. For the latter, as shown in the proof of Prop. 4.3 (Appx. G), this comes from the fact that for all shifts $y \in \mathcal{X}$,

$$\rho(x^{-1}(x(i) + y), x^{-1}(x(j) + y)) = \rho^P(x(j) + y - (x(i) + y)) = \rho^P(x(j) - x(i)) = \rho(i, j). \quad (12)$$

That is, from the fact that the relative positional encoding is invariant to the action of the translation group, i.e., $\mathcal{L}_y[\rho](i, j) = \rho(i, j), \ \forall y \in \mathcal{X}$. Similarly, the absence of positional encoding – more precisely, the use of a constant positional encoding –, is what allows for permutation equivariance

---

[3]$e$ represents a $0°$ rotation, i.e., the identity. The remaining elements $r^j$ represent rotations by $(90 \cdot j)°$.

(Prop. 4.1, Appx. G). Specifically, constant positional encodings $\rho_c(i) = c$, $\forall i \in \mathcal{S}$ are invariant to the action of the permutation group, i.e., $\mathcal{L}_\pi[\rho_c](i) = \rho_c(i)$, $\forall \pi \in \mathbb{S}_N$.

From these observations, we conclude that $\mathcal{G}$-equivariance is obtained by providing positional encodings which are ***invariant to the action of the group*** $\mathcal{G}$, i.e., s.t., $\mathcal{L}_g[\rho] = \rho$, $\forall g \in \mathcal{G}$. Furthermore, unique $\mathcal{G}$-equivariance is obtained by providing positional encodings which are invariant to the action of $\mathcal{G}$ but ***not invariant to the action of any other group*** $\mathcal{G}' \geq \mathcal{G}$. This is a key insight that allows us to provide (unique) equivariance to arbitrary symmetry groups, which we provide next.

## 5 GROUP EQUIVARIANT STAND-ALONE SELF-ATTENTION

In §4.3 we concluded that unique $\mathcal{G}$-equivariance is induced in self-attention by introducing positional encodings which are invariant to the action of $\mathcal{G}$ but not invariant to the action of other groups $\mathcal{G}' \geq \mathcal{G}$. However, this constraint does not provide any information about the expressivity of the mapping we have just made $\mathcal{G}$-equivariant. Let us first illustrate why this is important:

Consider the case of imposing rotation and translation equivariance to an encoding defined in $\mathbb{R}^2$. Since translation equivariance is desired, a relative positional encoding is required. For rotation equivariance, we must further impose the positional encoding to be equal for all rotations. That is $\mathcal{L}_\theta[\rho](i,j) \overset{!}{=} \rho(i,j)$, $\forall \theta \in [0, 2\pi]$, where $\mathcal{L}_\theta[\rho](i,j) := \rho^P(\theta^{-1}x(j) - \theta^{-1}x(i))$, and $\theta^{-1}$ depicts a rotation by $-\theta$ degrees. This constraint leads to an isotropic positional encoding unable to discriminate among orientations, which in turn enforces rotation invariance instead of rotation equivariance.[4] This is alleviated by *lifting* the underlying function on $\mathbb{R}^2$ to a space where rotations are explicitly encoded (Fig. B.1). To this end, one performs self-attention operations for positional encodings $\mathcal{L}_\theta[\rho]$ of varying values $\theta$ and indexes their responses by the corresponding $\theta$ value. Next, as rotations are now explicitly encoded, a positional encoding can be defined in this space which is able to discriminate among rotations (Fig. B.2). This in turn allows for rotation equivariance instead of rotation invariance.

It has been shown both theoretically (Ravanbakhsh, 2020) and empirically (Weiler & Cesa, 2019) that the most expressive class of $\mathcal{G}$-equivariant functions is given by functions that follow the regular representation of $\mathcal{G}$. In order to obtain feature representations that behave that way, we introduce a lifting self-attention layer (Fig. B.1, Eq. 14) that receives an input function on $\mathbb{R}^d$ and produces a feature representation on $\mathcal{G}$. Subsequently, arbitrarily many group self-attention layers (Fig. B.2, Eq. 16) interleaved with optional point-wise non-linearities can be applied. At the end of the network a feature representation on $\mathbb{R}^d$ can be provided by pooling over $\mathcal{H}$. In short, we provide a pure self-attention analogous to Cohen & Welling (2016). However, as the group acts directly on the positional encoding, our networks are *steerable* as well (Weiler et al., 2018b). This allows us to go beyond group discretizations that live in the grid without introducing interpolation artifacts (§5.1).

Though theoretically sound, neural architectures using regular representations are unable to handle continuous groups directly in practice. This is a result of the summation over elements $\tilde{h} \in \mathcal{H}$ in Eq. 15, which becomes an integral for continuous groups. Interestingly, using discrete groups does not seem to be detrimental in practice. Our experiments indicate that performance saturates for fine discrete approximations of the underlying continuous group (Tab. 2). In fact, (Weiler & Cesa, 2019, Tab. 3) show via extensive experiments that networks using regular representations and fine enough discrete approximations consistently outperform networks handling continuous groups via irreducible representations. We conjecture this is a result of the networks receiving discrete signals as input. As the action of several group elements fall within the same pixel, no further improvement can be obtained.

### 5.1 GROUP SELF-ATTENTION IS AN STEERABLE OPERATION

Convolutional filters are commonly parameterized by weights on a discrete grid, which approximate the function implicitly described by the filter at the grid locations. Unfortunately, for groups whose action does not live in this grid, e.g., 45° rotations, the filter must be interpolated. This is problematic as these filters are typically small and the resulting interpolation artifacts can be severe (Fig. 2a). *Steerable CNNs* tackle this problem by parameterizing convolutional filters on a continuous basis on which the action of the group is well-defined, e.g., *circular harmonics* (Weiler et al., 2018b), *B-splines*

---

[4]This phenomenon arises from the fact that $\mathbb{R}^2$ is a quotient of the roto-translation group. Consequently, imposing group equivariance in the quotient space is equivalent to imposing an additional homomorphism of constant value over its cosets. Conclusively, the resulting map is of constant value over the rotation elements and, thus, is not able to discriminate among them. See Ch. 3.1 of Dummit & Foote (2004) for an intuitive description.

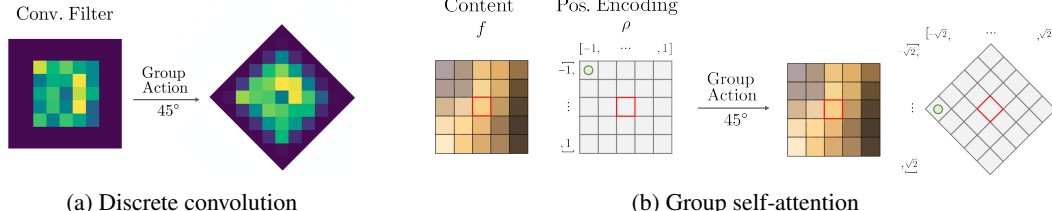

Figure 2: Steerability analysis of discrete convolutions and group self-attention.

(Bekkers, 2020). In group self-attention, the action of the group leaves the content of the image intact and only modifies the positional encoding (Figs. B.1, B.2). As the positional encoding lives on a continuous space, it can be transformed at an arbitrary grade of precision without interpolation (Fig. 2b).

## 5.2 LIFTING AND GROUP SELF-ATTENTION

**Lifting self-attention (Fig. B.1).** Let $\mathcal{G} = \mathbb{R}^d \rtimes \mathcal{H}$ be an affine group (Def. C.3) acting on $\mathbb{R}^d$. The *lifting self-attention* $m_{\mathcal{G}\uparrow}^r[f, \rho] : L_V(\mathbb{R}^d) \to L_{V'}(\mathcal{G})$ is a map from functions on $\mathbb{R}^d$ to functions on $\mathcal{G}$ obtained by modifying the relative positional encoding $\rho(i, j)$ by the action of group elements $h \in \mathcal{H}$: $\{\mathcal{L}_h[\rho](i, j)\}_{h \in \mathcal{H}}$, $\mathcal{L}_h[\rho](i, j) = \rho^P(h^{-1}x(j) - h^{-1}x(i))$. It corresponds to the concatenation of multiple self-attention operations (Eq. 11) indexed by $h$ with varying positional encodings $\mathcal{L}_h[\rho]$ :

$$m_{\mathcal{G}\uparrow}^r[f, \rho](i, h) = m^r[f, \mathcal{L}_h[\rho]](i) \tag{13}$$

$$= \varphi_{\text{out}}\Big( \bigcup_{h \in [H]} \sum_{j \in \mathcal{n}(i)} \sigma_j\big(\langle \varphi_{\text{qry}}^{(h)}(f(i)), \varphi_{\text{key}}^{(h)}(f(i) + \mathcal{L}_h[\rho](i, j)) \rangle\big) \varphi_{\text{val}}^{(h)}(f(j)) \Big). \tag{14}$$

**Proposition 5.1.** *Lifting self-attention is $\mathcal{G}$-equivariant. That is, it holds that:* $m_{\mathcal{G}\uparrow}^r[\mathcal{L}_g[f], \rho](i, h) = \mathcal{L}_g[m_{\mathcal{G}\uparrow}^r[f, \rho]](i, h)$.

**Group self-attention (Fig. B.2).** Let $\mathcal{G} = \mathbb{R}^d \rtimes \mathcal{H}$ be an affine group acting on itself and $f(i, \tilde{h}) \in L_V(\mathcal{G})$, $i \in \mathcal{S}$, $\tilde{h} \in \mathcal{H}$, be a function defined on a set immersed with the structure of the group $\mathcal{G}$. That is, enriched with a positional encoding $\rho((i, \tilde{h}), (j, \hat{h})) \coloneqq \rho^P((x(j) - x(i), \tilde{h}^{-1}\hat{h}))$, $i, j \in \mathcal{S}$, $\tilde{h}, \hat{h} \in \mathcal{H}$. The *group self-attention* $m_{\mathcal{G}}^r[f, \rho] : L_V(\mathcal{G}) \to L_{V'}(\mathcal{G})$ is a map from functions on $\mathcal{G}$ to functions on $\mathcal{G}$ obtained by modifying the group positional encoding by the action of group elements $h \in \mathcal{H}$: $\{\mathcal{L}_h[\rho]((i, \tilde{h}), (j, \hat{h}))\}_{h \in \mathcal{H}}$, $\mathcal{L}_h[\rho]((i, \tilde{h}), (j, \hat{h})) = \rho^P(h^{-1}(x(j) - x(i)), h^{-1}(\tilde{h}^{-1}\hat{h}))$. It corresponds to the concatenation of multiple self-attention operations (Eq. 11) indexed by $h$ with varying positional encodings $\mathcal{L}_h[\rho]$ and followed by a summation over the output domain along $\tilde{h}$:

$$m_{\mathcal{G}}^r[f, \rho](i, h) = \sum_{\tilde{h} \in \mathcal{H}} m^r[f, \mathcal{L}_h[\rho]](i, \tilde{h}) \tag{15}$$

$$= \varphi_{\text{out}}\Big( \bigcup_{h \in [H]} \sum_{\tilde{h} \in \mathcal{H}} \sum_{(j, \hat{h}) \in \mathcal{n}(i, \tilde{h})} \sigma_{j, \hat{h}}\big(\langle \varphi_{\text{qry}}^{(h)}(f(i, \tilde{h})), \varphi_{\text{key}}^{(h)}(f(j, \hat{h}) + \mathcal{L}_h[\rho]((i, \tilde{h}), (j, \hat{h}))) \rangle\big) \varphi_{\text{val}}^{(h)}(f(j, \hat{h})) \Big). \tag{16}$$

In contrast to vanilla and lifting self-attention, the group self-attention neighborhood $\mathcal{n}(i, \tilde{h})$ is now defined on the group. This allows distinguishing across group transformations, e.g., rotations.

**Proposition 5.2.** *Group self-attention is $\mathcal{G}$-equivariant. That is, it holds that:* $m_{\mathcal{G}}^r[\mathcal{L}_g[f], \rho](i, h) = \mathcal{L}_g[m_{\mathcal{G}}^r[f, \rho]](i, h)$.

Non-unimodular groups, i.e., groups that modify the volume of the objects they act upon, such as the dilation group, require a special treatment. This treatment is provided in Appx. E.

## 5.3 GROUP SELF-ATTENTION IS A GENERALIZATION OF THE GROUP CONVOLUTION

We have demonstrated that it is sufficient to define self-attention as a function on the group $\mathcal{G}$ and ensure that $\mathcal{L}_g[\rho] = \rho \; \forall g \in \mathcal{G}$ in order to enforce $\mathcal{G}$-equivariance. Interestingly, this observation is inline with the main statement of Kondor & Trivedi (2018) for (group) convolutions: "*the group convolution on $\mathcal{G}$ is the only (unique) $\mathcal{G}$-equivariant linear map*". In fact, our finding can be formulated as a generalization of Kondor & Trivedi (2018)'s statement as:

> **"*Linear mappings on $\mathcal{G}$ whose positional encoding is $\mathcal{G}$-invariant are $\mathcal{G}$-equivariant.*"**

This statement is more general than that of Kondor & Trivedi (2018), as it holds for data structures where (group) convolutions are not well-defined, e.g., sets, and it is equivalent to Kondor & Trivedi

(2018)'s statement for structures where (group) convolutions are well-defined. It is also congruent with results complementary to Kondor & Trivedi (2018) (Cohen et al., 2019a; Bekkers, 2020) as well as several works on group equivariance handling set-like structures like point-clouds (Thomas et al., 2018; Defferrard et al., 2020; Finzi et al., 2020; Fuchs et al., 2020) and symmetric sets (Maron et al., 2020).

In addition, we can characterize the expressivity of group self-attention. It holds that (*i*) group self-attention generalizes the group convolution and (*ii*) regular global group self-attention is an equivariant universal approximator. Statement (*i*) follows from the fact that any convolutional layer can be described as a multi-head self-attention layer provided enough heads (Cordonnier et al., 2020), yet self-attention often uses larger receptive fields. As a result, self-attention is able to describe a larger set of functions than convolutions, e.g., Fig. A.1. Cordonnier et al. (2020)'s statement can be seamlessly extended to group self-attention by incorporating an additional dimension corresponding to $\mathcal{H}$ in their derivations, and defining neighborhoods in this new space with a proportionally larger number of heads. Statement (*ii*) stems from the finding of Ravanbakhsh (2020) that functions induced by regular group representations are equivariant universal approximators provided *full kernels*, i.e., global receptive fields. Global receptive fields are required to guarantee that the equivariant map is able to model any dependency among input components. Global receptive fields are readily utilized by our proposed regular global group self-attention and, provided enough heads, one can ensure that any such dependencies is properly modelled.

## 6 EXPERIMENTS

We perform experiments on three image benchmark datasets for which particular forms of equivariance are desirable.[5] We evaluate our approach by contrasting `GSA-Nets` equivariant to multiple symmetry groups. Additionally, we conduct an study on rotMNIST to evaluate the performance of `GSA-Nets` as a function of the neighborhood size. All our networks follow the structure shown in Fig. F.1 and vary only in the number of blocks and channels. We emphasize that both the architecture and the number of parameters in `GSA-Nets` is left unchanged regardless of the group used. Our results illustrate that `GSA-Nets` consistently outperform equivalent non-equivariant attention networks. We further compare `GSA-Nets` with convolutional architectures. Though our approach does not build upon these networks, this comparison provides a fair view to the yet present gap between self-attention and convolutional architectures in vision tasks, also present in their group equivariant counterparts.

***Efficient implementation of lifting and group self-attention.*** Our self-attention implementation takes advantage of the fact that the group action only affects the positional encoding $\mathbf{P}$ to reduce the total computational cost of the operation. Specifically, we calculate self-attention scores w.r.t. the content $\mathbf{X}$ once, and reuse them for all transformed versions of the positional encoding $\{\mathcal{L}_{\hbar}[\rho]\}_{\hbar \in \mathcal{H}}$.

***Model designation.*** We refer to translation equivariant self-attention models as `Z2_SA`. Reflection equivariant models receive the keyword `M`, e.g., `Z2M_SA`, and rotation equivariant models the keyword `Rn`, where `n` depicts the angle discretization. For example, `R8_SA` depicts a model equivariant to rotations by 45 degrees. Specific model architectures are provided in Appx. F.

**RotMNIST.** The rotated MNIST dataset (Larochelle et al., 2007) is a classification dataset often used as a standard benchmark for rotation equivariance. It consists of $62k$ gray-scale 28x28 uniformly rotated handwritten digits, divided into training, validation and test sets of $10k$, $2k$ and 50k images. First, we study the effect of the neighborhood size on classification accuracy and convergence time. We train `R4_SA` networks for 300 epochs with vicinities `NxN` of varying size (Tab. 1, Fig. 3). Since `GSA-Nets` optimize where to attend, the complexity of the optimization problem grows as a function of `N`. Consequently, models with big vicinities are expected to converge slower. However, as the family of functions describable by big vicinities contains those describable by small ones, models with big vicinities are expected to be at least as good upon convergence. Our results show that models with small vicinities do converge much faster (Fig. 3). However, though some models with large vicinities do outperform models with small ones, e.g., `7x7` vs. `3x3`, a trend of this behavior is not apparent. We conjecture that 300 epochs are insufficient for all models to converge equally well. Unfortunately, due to computational constraints, we were not able to perform this experiment for a larger number of epochs. We consider an in-depth study of this behavior an important direction for future work.

Next, we compare `GSA-Nets` equivariant to translation and rotation at different angle discretizations (Tab. 2). Based on the results of the previous study, we select a `5x5` neighborhood, as it provides the

---

[5]Our code is publicly available at https://github.com/dwromero/g_selfatt.

Table 1: Accuracy vs. neighborhood size.

| ROTMNIST | | | |
|---|---|---|---|
| MODEL | NBHD. SIZE | ACC. (%) | TRAIN. TIME / EPOCH |
| R4_SA | 3x3 | 96.56 | 04:53 - 1GPU |
| | 5x5 | **97.49** | 05:34 - 1GPU |
| | 7x7 | 97.33 | 09:03 - 1GPU |
| | 9x9 | 97.42 | 09:16 - 1GPU |
| | 11x11 | 97.17 | 12:09 - 1GPU |
| | 15x15 | 96.89 | 10:27 - 2GPU |
| | 19x19 | 96.86 | 14:27 - 2GPU |
| | 23x23 | 97.05 | 06:13 - 3GPU |
| | 28x28 | 96.81 | 12:12 - 4GPU |

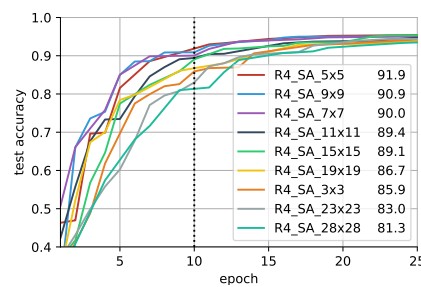

Figure 3: Test accuracy in early training stage.

Table 2: Classification results. All convolutional architectures use `3x3` filters.

| ROTMNIST | | | CIFAR10 | | | PATCHCAMELYON | | |
|---|---|---|---|---|---|---|---|---|
| MODEL | ACC. (%) | PARAMS. | MODEL | ACC. (%) | PARAMS. | MODEL | ACC. (%) | PARAMS. |
| Z2_SA | 96.37 | | Z2_SA | 82.3 | 2.99M | Z2_SA | 83.04 | 205.66K |
| R4_SA | 97.46 | | Z2M_SA | **83.72** | | R4_SA | 83.44 | |
| R8_SA | 97.90 | 44.67K | | | | R8_SA | 83.58 | |
| R12_SA | **97.97** | | Z2_CNN[+] | **90.56** | 1.37M | R4M_SA | **84.76** | |
| R16_SA | 97.66 | | [+]Cohen & Welling (2016). | | | | | |
| Z2_CNN[+] | 94.97 | 21.75K | | | | Z2_CNN[†] | 84.07 | 130.60K |
| R4_CNN[†] | 98.21 | 77.54K | | | | R4_CNN[†] | 87.55 | 129.65K |
| α-R4_CNN[†] | **98.31** | 73.13K | | | | R4M_CNN[†] | 88.36 | 124.21K |
| [+]Cohen & Welling (2016) | | | | | | $\alpha_F$-R4_CNN[†] | 88.66 | 140.45K |
| [†]Romero et al. (2020a) | | | | | | $\alpha_F$-R4M_CNN[†] | **89.12** | 141.22K |
| | | | | | | [†]Romero et al. (2020a). | | |

best trade-off between accuracy and convergence time. Our results show that finer discretizations lead to better accuracy but saturates around `R12`. We conjecture that this is due to the discrete resolution of the images in the dataset, which leads finer angle discretizations to fall within the same pixel.

**CIFAR-10.** The CIFAR-10 dataset (Krizhevsky et al., 2009) consists of $60k$ real-world 32x32 RGB images uniformly drawn from 10 classes, divided into training, validation and test sets of $40k$, $10k$ and $10k$ images. Since reflection is a symmetry that appears ubiquitously in natural images, we compare `GSA-Nets` equivariant to translation and reflection in this dataset (Tab. 2). Our results show that reflection equivariance indeed improves the classification performance of the model.

**PCam.** The PatchCamelyon dataset (Veeling et al., 2018) consists of $327k$ 96x96 RGB image patches of tumorous/non-tumorous breast tissues extracted from Camelyon16 (Bejnordi et al., 2017). Each patch is labeled as tumorous if the central region (32x32) contains at least one tumour pixel. As cells appear at arbitrary positions and poses, we compare `GSA-Nets` equivariant to translation, rotation and reflection (Tab. 2). Our results show that incorporating equivariance to reflection in addition to rotation, as well as providing finer group discretization, improve classification performance.

## 7 DISCUSSION AND FUTURE WORK

Though `GSA-Nets` perform competitively to `G-CNNs` for some tasks, `G-CNNs` still outperforms our approach in general. We conjecture that this is due to the harder nature of the optimization problem in `GSA-Nets` and the carefully crafted architecture design, initialization and optimization procedures developed for CNNs over the years. Though our theoretical results indicate that `GSA-Nets` can be more expressive than `G-CNNs` (§ 5.3), further research in terms of design, optimization, stability and generalization is required. These are in fact open questions for self-attention in general (Xiong et al., 2020; Liu et al., 2020; Zhao et al., 2020) and developments in this direction are of utmost importance.

The main drawback of our approach is the quadratic memory and time complexity typical of self-attention. This is an active area of research, e.g., Kitaev et al. (2020); Wang et al. (2020); Zaheer et al. (2020); Choromanski et al. (2020) and we believe that efficiency advances to vanilla self-attention can be seamlessly integrated in `GSA-Nets`. Our theoretical results indicate that `GSA-Nets` have the potential to become the standard solution for applications exhibiting symmetries, e.g., medical imagery. In addition, as self-attention is a set operation, `GSA-Nets` provide straightforward solutions to set-like data types, e.g., point-clouds, graphs, symmetric sets, which may benefit from additional geometrical information, e.g., Fuchs et al. (2020); Maron et al. (2020). Finally, we hope our theoretical insights serve as a support point to further explore and understand the construction of equivariant maps for graphs and sets, which often come equipped with spatial coordinates: a type of positional encoding.

## ACKNOWLEDGMENTS

We gratefully acknowledge Michael Hutchinson, Charline Le Lan, Sheheryar Zaidi, Emilien Dupont, and Hyunjik Kim for useful discussions, Robert-Jan Bruintjes, Fabian Fuchs, Erik Bekkers, Andreas Loukas, Mark Hoogendoorn and our anonymous reviewers for their valuable comments on early versions of this work which largely helped us to improve the quality of our work.

David W. Romero is financed as part of the Efficient Deep Learning (EDL) programme (grant number P16-25), partly funded by the Dutch Research Council (NWO) and Semiotic Labs. Jean-Baptiste Cordonnier is financed by the Swiss Data Science Center (SDSC). Both authors thankfully acknowledge everyone involved in funding this work. This work was carried out on the Dutch national e-infrastructure with the support of SURF Cooperative.

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

# APPENDIX

## A CONVOLUTION AND THE SELF-ATTENTION: A GRAPHICAL COMPARISON

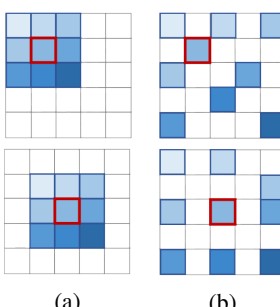

(a)                (b)

Figure A.1: Parameter usage in convolutional kernels (Fig. A.1a) and self-attention (Fig. A.1b). Given a budget of 9 parameters, a convolutional filter ties these parameters to specific positions. Subsequently, these parameters remain static regardless of (*i*) the query input position and (*ii*) the input signal itself. Self-attention, on the other hand, does not tie parameters to any specific positions at all. Contrarily, it compares the representations of all tokens falling in its receptive field. As a result, provided enough heads, self-attention is more general than convolutions, as it can represent any convolutional kernel, e.g., Fig. A.1a, as well as several other functions defined on its receptive field.

## B LIFTING AND GROUP SELF-ATTENTION: A GRAPHICAL DESCRIPTION

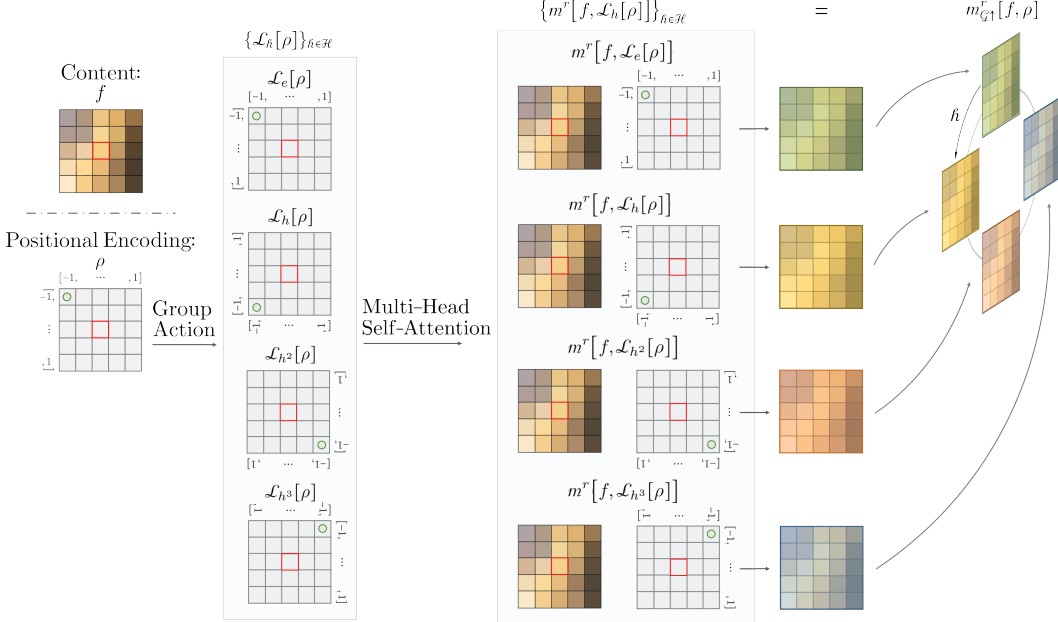

Figure B.1: Lifting self-attention on the roto-translation group for discrete rotations by 90 degrees (also called the $\mathbb{Z}_4$ group). The $\mathbb{Z}_4$ group is defined as $\bar{\mathcal{H}} = \{e, h, h^2, h^3\}$, where $h$ depicts a 90° rotation. The lifting self-attention corresponds to the concatenation of $|\mathcal{H}| = 4$ self-attention operations between the input $f$ and $h$-transformed versions of the positional encoding $\mathcal{L}[\rho]$, $\forall h \in \mathcal{H}$. As a result, the model "sees" the input $f$ at each of the rotations in the group at once. Since $\mathbb{Z}_4$ is a cyclic group, i.e., $h^4 = e$, functions on this group are often represented as responses on a ring (right side of the image). This is a self-attention analogous to the regular lifting group convolution broadly utilized in group equivariant learning literature, e.g., Cohen & Welling (2016); Romero et al. (2020a).

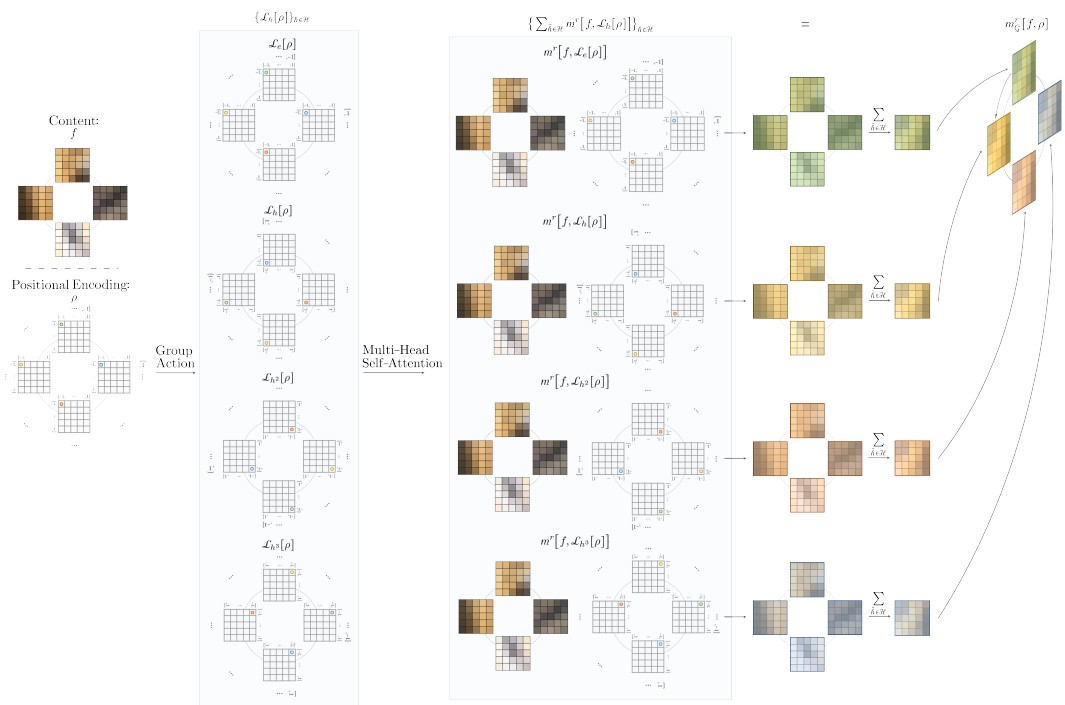

Figure B.2: Lifting self-attention on the roto-translation group for discrete rotations by 90 degrees (also called the $\mathbb{Z}_4$ group). The $\mathbb{Z}_4$ group is defined as $\mathcal{H} = \{e, \hbar, \hbar^2, \hbar^3\}$, where $\hbar$ depicts a 90° rotation. Analogous to lifting self-attention (Fig. B.1), group self-attention corresponds to a concatenation of $|\mathcal{H}| = 4$ self-attention operations between the input $f$ and $\hbar$-transformed versions of the positional encoding $\mathcal{L}[\rho]$, $\forall \hbar \in \mathcal{H}$. However, in contrast to lifting self-attention, both $f$ and $\rho$ are now defined on the group $\mathcal{G}$. Consequently, an additional sum over $\tilde{\hbar}$ is required during the operation (c.f., Eq. 16). Since $\mathbb{Z}_4$ is a cyclic group, i.e., $\hbar^4 = e$, functions on $\mathbb{Z}_4$ are often represented as responses on a ring (right side of the image). This is a self-attention analogous to the regular group convolution broadly utilized in group equivariant learning literature, e.g., Cohen & Welling (2016); Romero et al. (2020a).

## C  CONCEPTS FROM GROUP THEORY

**Definition C.1 (Group).** *A group is an ordered pair $(\mathcal{G}, \cdot)$ where $\mathcal{G}$ is a set and $\cdot : \mathcal{G} \times \mathcal{G} \to \mathcal{G}$ is a binary operation on $\mathcal{G}$, such that* (i) *the set is closed under this operation,* (ii) *the operation is associative, i.e., $(g_1 \cdot g_2) \cdot g_3 = g_1 \cdot (g_2 \cdot g_3)$, $g_1, g_2, g_3 \in \mathcal{G}$,* (iii) *there exists an identity element $e \in \mathcal{G}$ s.t. $\forall g \in \mathcal{G}$ we have $e \cdot g = g \cdot e = g$, and* (iv) *for each $g \in \mathcal{G}$, there exists an inverse $g^{-1}$ s.t. $g \cdot g^{-1} = e$.*

**Definition C.2 (Subgroup).** *Let $(\mathcal{G}, \cdot)$ be a group. A subset $\mathcal{H}$ of $\mathcal{G}$ is a subgroup of $\mathcal{G}$ if $\mathcal{H}$ is nonempty and closed under the group operation and inverses (i.e., $\hbar_1, \hbar_2 \in \mathcal{H}$ implies that $\hbar_1^{-1} \in \mathcal{H}$ and $\hbar_1 \cdot \hbar_2 \in \mathcal{H}$). If $\mathcal{H}$ is a subgroup of $\mathcal{G}$ we write $\mathcal{H} \le \mathcal{G}$*

**Definition C.3 (Semi-direct product and affine groups).** *In practice, one is mainly interested in the analysis of data defined on $\mathbb{R}^d$, and, consequently, in groups of the form $\mathcal{G} = \mathbb{R}^d \rtimes \mathcal{H}$, resulting from the semi-direct product ($\rtimes$) between the translation group ($\mathbb{R}^d$, $+$) and an arbitrary (Lie) group $\mathcal{H}$ that acts on $\mathbb{R}^d$, e.g., rotation, scaling, mirroring, etc. This family of groups is referred to as* affine groups *and their group product is defined as:*

$$g_1 \cdot g_2 = (x_1, \hbar_1) \cdot (x_2, \hbar_2) = (x_1 + \hbar_1 \odot x_2, \hbar_1 \cdot \hbar_2), \tag{17}$$

*with $g = (x_1, \hbar_1)$, $g_2 = (x_2, \hbar_2) \in G$, $x_1, x_2 \in \mathbb{R}^d$ and $\hbar_1, \hbar_2 \in \mathcal{H}$. The operator $\odot$ denotes the action of $\hbar \in \mathcal{H}$ on $x \in \mathbb{R}^d$, and it describes how a vector $x \in \mathbb{R}^d$ is modified by elements $\hbar \in \mathcal{H}$.*

**Definition C.4 (Group representation).** *Let $\mathcal{G}$ be a group and $\mathbb{L}^2(X)$ be a space of functions defined on some vector space $X$. The (left) regular group representation of $\mathcal{G}$ is a linear transformation $\mathcal{L} : \mathcal{G} \times \mathbb{L}^2(X) \to \mathbb{L}^2(X)$, $(g, f) \mapsto \mathcal{L}_g[f] := f(g^{-1} \odot x)$, that shares the group structure via:*

$$\mathcal{L}_{g_1} \mathcal{L}_{g_2}[f] = \mathcal{L}_{g_1 \cdot g_2}[f] \tag{18}$$

*for any $g_1, g_2 \in G$, $f \in \mathbb{L}_2(X)$. That is, concatenating two such transformations, parameterized by*

*$g_1$ and $g_2$, is equivalent to a single transformation parameterized by $g_1 \cdot g_2 \in \mathcal{G}$. If the group $\mathcal{G}$ is affine, the group representation $\mathcal{L}_g$ can be split as:*

$$\mathcal{L}_g[f] = \mathcal{L}_x \mathcal{L}_\hbar[f], \tag{19}$$

*with $g = (x, \hbar) \in \mathcal{G}$, $x \in \mathbb{R}^d$ and $\hbar \in \mathcal{H}$. Intuitively, the representation of $\mathcal{G}$ on a function $f$ describes how the function as a whole, i.e., $f(x), \forall x \in X$, is transformed by the effect of group elements $g \in \mathcal{G}$.*

## D  Actions and Representations of Groups Acting on Homogeneous Spaces for Functions Defined on Sets

In this section we show that the action of a group $\mathcal{G}$ acting on a homogeneous space $\mathcal{X}$ is well defined on sets $\mathcal{S}$ gathered from $\mathcal{X}$, and that it induces a group representation of functions defined on $\mathcal{S}$.

Let $\mathcal{S} = \{i\}$ be a set and $\mathcal{X}$ be a homogeneous space on which the action of $\mathcal{G}$ is well-defined, i.e., $gx \in \mathcal{X}, \forall g \in \mathcal{G}, \forall x \in \mathcal{X}$. Since $\mathcal{S}$ has been gathered from $\mathcal{X}$, there exists an injective map $x : \mathcal{S} \to \mathcal{X}$, that maps set elements $i \in \mathcal{S}$ to unique elements $x_i \in \mathcal{X}$. That is, there exists a map $x : i \mapsto x_i$ that assigns an unique value $x_i \in \mathcal{X}$ to each $i \in \mathcal{S}$ corresponding to the coordinates from which the set element has been gathered.

Since the action of $\mathcal{G}$ is well defined on $\mathcal{X}$, it follows that the left regular representation (Def. C.4) $\mathcal{L}_g[f^{\mathcal{X}}](x_i) \coloneqq f^{\mathcal{X}}(g^{-1}x_i) \in L_y(\mathcal{X})$ of functions $f^{\mathcal{X}} \in L_y(\mathcal{X})$ exists and is well-defined. Since $x$ is injective, the left regular representation $\mathcal{L}_g[f^{\mathcal{X}}](x_i) = f^{\mathcal{X}}(g^{-1}x_i)$ can be expressed uniquely in terms of set indices as $\mathcal{L}_g[f^{\mathcal{X}}](x_i) = f^{\mathcal{X}}(g^{-1}x_i) = f^{\mathcal{X}}(g^{-1}x(i))$. Furthermore, its inverse $x^{-1} : \mathcal{X} \to \mathcal{S}$, $x^{-1} : x_i \to i$ also exist and is well-defined. As a consequence, points $x_i \in \mathcal{X}$ can be expressed uniquely in terms of set indices as $i = x^{-1}(x_i), i \in \mathcal{S}$. Consequently, functions $f^{\mathcal{X}} \in L_y(\mathcal{X})$ can be expressed in terms of functions $f \in L_y(\mathcal{S})$ by means of the equality $f^{\mathcal{X}}(i) = f(x^{-1}(x_i))$. Resultantly, we see that the group representation $\mathcal{L}_g[f^{\mathcal{X}}](x_i) = f^{\mathcal{X}}(g^{-1}x(i))$ can be described in terms of functions $f \in L_y(\mathcal{S})$ as:

$$\mathcal{L}_g[f^{\mathcal{X}}](x_i) = f^{\mathcal{X}}(g^{-1}(x_i)) = f^{\mathcal{X}}(g^{-1}x(i)) = f(x^{-1}(g^{-1}x(i))) = \mathcal{L}_g[f](i),$$

with a corresponding group representation on $L_y(\mathcal{S})$ given by $\mathcal{L}_g[f](i) = f(x^{-1}(g^{-1}x(i)))$, and an action of group elements $g \in \mathcal{G}$ on set elements $i$ given by $gi \coloneqq x^{-1}(gx(i))$.

## E  The Case of Non-Unimodular Groups: Self-Attention on the Dilation-Translation Group

The lifting and group self-attention formulation provided in §5.2 are only valid for unimodular groups. That is, for groups whose action does not change the volume of the objects they act upon, e.g., rotation, mirroring, etc. Non-unimodular groups, however, do modify the volume of the acted objects (Bekkers, 2020). The most relevant non-unimodular group for this work is the dilation group $\mathcal{H} = (\mathbb{R}_{>0}, \times)$. To illustrate why this distinction is important, consider the following example:

Imagine we have a circle on $\mathbb{R}^2$ of area $\pi r^2$. If we rotate, mirror or translate the circle, its size is kept constant. If we increase its radius by a factor $\hbar \in \mathbb{R}_{>0}$, however, its size would increase by $\hbar^2$. Imagine that we have an application for which we would like to recognize this circle regardless of any of these transformations by means of self-attention. For this purpose, we define a neighborhood $\mathcal{N}$ for which the original circle fits perfectly. Since the size of the circle is not modified for any translated, rotated or translated versions of it, we would still be able to detect the circle regardless of these transformations. If we scale the circle by a factor of $\hbar > 1$, however, the circle would fall outside of our neighborhood $\mathcal{N}$ and hence, we would not be able to recognize it.

A solution to this problem is to scale our neighborhood $\mathcal{N}$ in a proportional way. That is, if the circle is scaled by a factor $\hbar \in \mathbb{R}_{>0}$, we scale our neighborhood by the same factor $\hbar \colon \mathcal{N} \to \hbar\mathcal{N}$. Resultantly, the circle would fall within the neighborhood for any scale factor $\hbar \in \mathbb{R}_{>0}$. Unfortunately, there is a problem: self-attention utilizes summations over its neighborhood. Since $\sum_{i \in \hbar\mathcal{N}} i > \sum_{i \in \mathcal{N}} i$, for $\hbar > 1$, and $\sum_{i \in \hbar\mathcal{N}} i < \sum_{i \in \mathcal{N}} i$, for $\hbar < 1$, the result of the summations would still differ for different scales. Specifically, this result would always be bigger for larger versions of the neighborhood. This is problematic, as the response produced by the same circle, would still be different for different scales.

In order to handle this problem, one utilizes a normalization factor proportional to the change of size of the neighborhood considered. This ensures that the responses are equivalent for any scale $\hbar \in \mathbb{R}_{>0}$. That is, one normalizes all summations proportionally to the size of the neighborhood. As a result, we obtain that $\sum_{i \in \hbar_1} n(\hbar_1^2)^{-1} i = \sum_{i \in \hbar_2} n(\hbar_2^2)^{-1} i, \; \forall \hbar_1, \hbar_2 \in \mathbb{R}_{>0}$.[6]

In the example above we have provided an intuitive description of the (left invariant) Haar measure $\mathrm{d}\mu(\hbar)$. As its name indicates, it is a measure defined on the group, which is invariant over all group elements $\hbar \in \mathcal{H}$. For several unimodular groups, the Haar measure corresponds to the Lebesgue measure as the volume of the objects the group acts upon is kept equal, i.e., $\mathrm{d}\mu(\hbar) = \mathrm{d}\hbar$.[7] For non-unimodular groups, however, the Haar measure requires a normalization factor proportional to the change of volume of these objects. Specifically, the Haar measure corresponds to the Lebesgue measure times a normalization factor $\hbar^d$, where $d$ corresponds to the dimensionality of the space $\mathbb{R}^d$ the group acts upon (Bekkers, 2020; Romero et al., 2020b), i.e., $\mathrm{d}\mu(\hbar) = \frac{1}{\hbar^d} \mathrm{d}\hbar$.

In conclusion, in order to obtain group equivariance to non-unimodular groups, the lifting and group self-attention formulation provided in Eqs. 14, 16 must be modified via normalization factors proportional to the group elements $\hbar \in \mathcal{H}$. Specifically, they are redefined as:

$$m_{\mathcal{G}\uparrow}^r[f,\rho](i,\hbar) = \varphi_{\mathrm{out}}\Big( \bigcup_{h \in [H]} \sum_{j \in \hbar \mathcal{N}(i)} \tfrac{1}{\hbar^d} \, \sigma_j\big(\langle \varphi_{\mathrm{qry}}^{(h)}(f(i)), \varphi_{\mathrm{key}}^{(h)}(f(i) + \mathcal{L}_\hbar[\rho](i,j))\rangle\big) \varphi_{\mathrm{val}}^{(h)}(f(j))\Big) \quad (20)$$

$$m_{\mathcal{G}}^r[f,\rho](i,\hbar) = \varphi_{\mathrm{out}}\Big( \bigcup_{h \in [H]} \sum_{\tilde{h} \in \mathcal{H}} \sum_{(j,\hat{h}) \in \hbar \mathcal{N}(i,\tilde{h})} \tfrac{1}{\hbar^{d+1}} \, \sigma_{j,\hat{h}}\big(\langle \varphi_{\mathrm{qry}}^{(h)}(f(i,\tilde{h})), \varphi_{\mathrm{key}}^{(h)}(f(j,\hat{h}) \\ + \mathcal{L}_\hbar[\rho]((i,\tilde{h}),(j,\hat{h}))\rangle\big) \varphi_{\mathrm{val}}^{(h)}(f(j,\hat{h}))\Big). \quad (21)$$

The factor $d+1$ in Eq. 21 results from the fact that the summation is now performed on the group $\mathcal{G} = \mathbb{R}^d \rtimes \mathcal{H}$, an space of dimensionality $d+1$. An interesting case emerges when global neighborhoods are considered, i.e., s.t. $\mathcal{N}(i) = \mathcal{S}, \; \forall i \in \mathcal{S}$. Since $\hbar \mathcal{N}(i) = \mathcal{N}(i) = \mathcal{S}$ for any $\hbar > 1$, approximation artifacts are introduced. It is not clear if it is better to introduce normalization factors in these situations or not. An in-depth investigation of this phenomenon is left for future research.

### E.1 CURRENT EMPIRICAL ASPECTS OF SCALE EQUIVARIANT SELF-ATTENTION

Self-attention suffers from quadratic memory and time complexity proportional to the size of the neighborhood considered. This constraint is particularly important for the dilation group, for which these neighborhoods grow as a result of the group action. We envisage two possible solutions to this limitation left out for future research:

The most promising solution is given by incorporating recent advances in efficient self-attention in group self-attention, e.g., Kitaev et al. (2020); Wang et al. (2020); Zaheer et al. (2020); Katharopoulos et al. (2020); Choromanski et al. (2020). By reducing the quadratic complexity of self-attention, the current computational constraints of scale equivariant self-attention can be (strongly) reduced. Importantly, resulting architectures would be comparable to Bekkers (2020); Sosnovik et al. (2020); Romero et al. (2020b) in terms of their functionality and the group discretizations they can manage.

The second option is to draw a self-attention analogous to Worrall & Welling (2019), where scale equivariance is implemented via dilated convolutions. One might consider an analogous to dilated convolutions via "sparse" dilations of the self-attention neighborhood. As a result, scale equivariance can be implemented while retaining an equal computational cost for all group elements. Importantly however, this strategy is viable for a dyadic set of scales only, i.e., a set of scales given by a set $\{2^j\}_{j=0}^{j_{\max}}$, and thus, less general that the scale-equivariant architectures listed before.

## F EXPERIMENTAL DETAILS

In this section we provide extended details over our implementation as well as the exact architectures and optimization schemes used in our experiments. All our models follow the structure shown in Fig. F.1 and vary only in the number of blocks and channels. All self-attention operations utilize 9 heads. We utilize `PyTorch` for our implementation. Any missing specification can be safely considered to be the `PyTorch` default value. Our code is publicly available at https://github.com/dwromero/g_selfatt.

---

[6] The squared factor in $\hbar_1^2$ and $\hbar_2^2$ appears as a result that the neighborhood growth is quadratic in $\mathbb{R}^2$.

[7] This is why this subtlety is often left out in group equivariance literature.

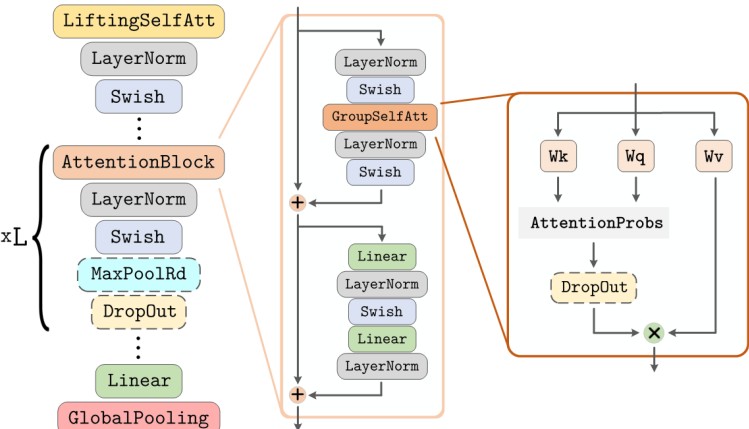

Figure F.1: Graphical description of group self-attention networks. Dot-lined blocks depict optional blocks. Linear layers are applied point-wise across the feature map. Swish non-linearities (Ramachandran et al., 2017) and layer normalization (Ba et al., 2016) are used all across the network. The `GlobalPooling` block consists of max-pool over group elements followed by spatial mean-pool.

### F.1 ROTATED MNIST

For rotational MNIST we use a group self-attention network composed of 5 attention blocks with 20 channels. We utilize automatic mixed precision during training to reduce memory requirements. `attention_dropout_rate` and `value_dropout_rate` are both set to 0.1. We train for 300 epochs and utilize the Adam optimizer, batch size of 8, weight decay of 0.0001 and learning rate of 0.001.

### F.2 CIFAR-10

For CIFAR-10 we use a group self-attention network composed of 6 attention blocks with 96 channels for the first two blocks and 192 channels for the rest. `attention_dropout_rate` and `value_dropout_rate` are both set to 0.1. We use dropout on the input with a rate of 0.3 and additional dropout blocks of rate 0.2 followed by spatial max-pooling after the second and fourth block. We did not use automatic mixed precision training for this dataset as it made all models diverge. We perform training for 350 epochs and utilize stochastic gradient descent with a momentum of 0.9 and cosine learning rate scheduler with base learning rate 0.01 (Loshchilov & Hutter, 2016). We utilize a batch size of 24, weight decay of 0.0001 and He's initialization.

### F.3 PATCHCAMELYON

For PatchCamelyon we use a group self-attention network composed of 4 attention blocks with 12 channels for the first block, 24 channels for the second block, 48 channels for the third and fourth blocks and 96 channels for the last block. `attention_dropout_rate` and `value_dropout_rate` are both set to 0.1. We use an additional max-pooling block after the lifting block to reduce memory requirements. We did not use automatic mixed precision training for this dataset as it made all models diverge. We perform training for 100 epochs, utilize stochastic gradient descent with a momentum of 0.9 and cosine learning rate scheduler with base learning rate 0.01 (Loshchilov & Hutter, 2016). We utilize a batch size of 8, weight decay of 0.0001 and He's initialization.

## G PROOFS

***Proof of Proposition 4.1.*** If the self-attention formulation provided in Eqs. 3, 8, is permutation equivariant, then it must hold that $m[\mathcal{L}_\pi[f]](i) = \mathcal{L}_\pi[m[f]](i)$. Consider a permuted input signal $\mathcal{L}_\pi[f](i) = f(\pi^{-1}(i))$. The self-attention operation on $\mathcal{L}_\pi[f]$ is given by:

$$m\big[\mathcal{L}_\pi[f]\big](i) = \varphi_{\text{out}}\Big(\bigcup_{h\in[H]}\sum_{j\in\mathcal{S}}\sigma_j\big(\langle\varphi_{\text{qry}}^{(h)}(\mathcal{L}_\pi[f](i)),\varphi_{\text{key}}^{(h)}(\mathcal{L}_\pi[f](j))\rangle\big)\varphi_{\text{val}}^{(h)}(\mathcal{L}_\pi[f](j))\Big)$$

$$= \varphi_{\text{out}}\Big(\bigcup_{h\in[H]}\sum_{j\in\mathcal{S}}\sigma_j\big(\langle\varphi_{\text{qry}}^{(h)}(f(\pi^{-1}(i))),\varphi_{\text{key}}^{(h)}(f(\pi^{-1}(j)))\rangle\big)\varphi_{\text{val}}^{(h)}(f(\pi^{-1}(j)))\Big)$$

$$= \varphi_{\text{out}}\Big( \bigcup_{h \in [H]} \sum_{\pi(\bar{j}) \in \mathcal{S}} \sigma_{\pi(\bar{j})}\big( \langle \varphi_{\text{qry}}^{(h)}(f(\bar{\imath})), \varphi_{\text{key}}^{(h)}(f(\bar{j})) \rangle \big) \varphi_{\text{val}}^{(h)}(f(\bar{j})) \Big)$$

$$= \varphi_{\text{out}}\Big( \bigcup_{h \in [H]} \sum_{\bar{j} \in \mathcal{S}} \sigma_{\bar{j}}\big( \langle \varphi_{\text{qry}}^{(h)}(f(\bar{\imath})), \varphi_{\text{key}}^{(h)}(f(\bar{j})) \rangle \big) \varphi_{\text{val}}^{(h)}(f(\bar{j})) \Big)$$

$$= m[f](\bar{\imath}) = m[f](\pi^{-1}(i))$$

$$= \mathcal{L}_{\pi}\big[ m[f] \big](i)$$

Here we have used the substitution $\bar{\imath} = \pi(i)$ and $\bar{j} = \pi(j)$. Since the summation is defined over the entire set we have that $\sum_{\pi(\bar{j}) \in \mathcal{S}}[\cdot] = \sum_{\bar{j} \in \mathcal{S}}[\cdot]$. Conclusively, we see that $m[\mathcal{L}_{\pi}[f]](i) = \mathcal{L}_{\pi}[m[f]](i)$. Hence, permutation equivariance indeed holds. □

***Proof of Claim 4.2. Permutation equivariance.*** If the self-attention formulation provided in Eq. 10 is permutation equivariant, then it must hold that $m[\mathcal{L}_{\pi}[f], \rho](i) = \mathcal{L}_{\pi}[m[f, \rho]](i)$. Consider a permuted input signal $\mathcal{L}_{\pi}[f](i) = f(\pi^{-1}(i))$. The self-attention operation on $\mathcal{L}_{\pi}[f]$ is given by:

$$m\big[ \mathcal{L}_{\pi}[f], \rho \big](i)$$
$$= \varphi_{\text{out}}\Big( \bigcup_{h \in [H]} \sum_{j \in \mathcal{N}(i)} \sigma_j\big( \langle \varphi_{\text{qry}}^{(h)}(\mathcal{L}_{\pi}[f](i) + \rho(i)), \varphi_{\text{key}}^{(h)}(\mathcal{L}_{\pi}[f](j) + \rho(j)) \rangle \big) \varphi_{\text{val}}^{(h)}(\mathcal{L}_{\pi}[f](j)) \Big)$$

As discussed in §3.2, since there exists permutations in $\mathbb{S}_N$ able to send elements $j$ in $\mathcal{N}(i)$ to elements $\tilde{j}$ outside of $\mathcal{N}(i)$, it is trivial to show that Eq. 10 is not equivariant to $\mathbb{S}_N$. Consequently, in order to provide a more interesting analysis, we consider global attention here, i.e., cases where $\mathcal{N}(i) = \mathcal{S}$. As shown for Proposition 4.1, this self-attention instantiation is permutation equivariant. Consequently, by considering this particular case, we are able to explicitly analyze the effect of introducing absolute positional encodings into the self-attention formulation. We have then that:

$$m\big[ \mathcal{L}_{\pi}[f], \rho \big](i)$$
$$= \varphi_{\text{out}}\Big( \bigcup_{h \in [H]} \sum_{j \in \mathcal{S}} \sigma_j\big( \langle \varphi_{\text{qry}}^{(h)}(f(\pi^{-1}(i)) + \rho(i)), \varphi_{\text{key}}^{(h)}(f(\pi^{-1}(j)) + \rho(j)) \rangle \big) \varphi_{\text{val}}^{(h)}(f(\pi^{-1}(j))) \Big)$$

$$= \varphi_{\text{out}}\Big( \bigcup_{h \in [H]} \sum_{\pi(\bar{j}) \in \mathcal{S}} \sigma_{\pi(\bar{j})}\big( \langle \varphi_{\text{qry}}^{(h)}(f(\bar{\imath}) + \rho(\pi(\bar{\imath}))), \varphi_{\text{key}}^{(h)}(f(\bar{j}) + \rho(\pi(\bar{j}))) \rangle \big) \varphi_{\text{val}}^{(h)}(f(\bar{j})) \Big)$$

$$= \varphi_{\text{out}}\Big( \bigcup_{h \in [H]} \sum_{\bar{j} \in \mathcal{S}} \sigma_{\bar{j}}\big( \langle \varphi_{\text{qry}}^{(h)}(f(\bar{\imath}) + \rho(\pi(\bar{\imath}))), \varphi_{\text{key}}^{(h)}(f(\bar{j}) + \rho(\pi(\bar{j}))) \rangle \big) \varphi_{\text{val}}^{(h)}(f(\bar{j})) \Big)$$

Here we have used the substitution $\bar{\imath} = \pi(i)$ and $\bar{j} = \pi(j)$. Since the summation is defined over the entire set we have that $\sum_{\pi(\bar{j}) \in \mathcal{S}}[\cdot] = \sum_{\bar{j} \in \mathcal{S}}[\cdot]$. Since $\rho(\pi(\bar{\imath})) \neq \rho(\bar{\imath})$ and $\rho(\pi(\bar{j})) \neq \rho(\bar{j})$, we are unable to reduce the expression further towards the form of $m[f, \rho](\bar{\imath})$. Consequently, we conclude that absolute position-aware self-attention is not permutation equivariant.

***Translation equivariance.*** If the self-attention formulation provided in Eq. 10, is translation equivariant, then it must hold that $m(\mathcal{L}_y[f], \rho)(i) = \mathcal{L}_y[m(f, \rho)](i)$. Consider a translated input signal $\mathcal{L}_y[f](i) = f(x^{-1}(x(i) - y))$. The self-attention operation on $\mathcal{L}_y[f]$, is given by:

$$m\big[ \mathcal{L}_y[f], \rho \big](i)$$
$$= \varphi_{\text{out}}\Big( \bigcup_{h \in [H]} \sum_{j \in \mathcal{N}(i)} \sigma_j\big( \langle \varphi_{\text{qry}}^{(h)}(\mathcal{L}_y[f](i) + \rho(i)), \varphi_{\text{key}}^{(h)}(\mathcal{L}_y[f](j) + \rho(j)) \rangle \big) \varphi_{\text{val}}^{(h)}(\mathcal{L}_y[f](j)) \Big)$$

$$= \varphi_{\text{out}}\Big( \bigcup_{h \in [H]} \sum_{j \in \mathcal{N}(i)} \sigma_j\big( \langle \varphi_{\text{qry}}^{(h)}(f(x^{-1}(x(i) - y)) + \rho(i)),$$
$$\varphi_{\text{key}}^{(h)}(f(x^{-1}(x(j) - y)) + \rho(j)) \rangle \big) \varphi_{\text{val}}^{(h)}(f(x^{-1}(x(j) - y))) \Big)$$

$$= \varphi_{\text{out}}\Big( \bigcup_{h \in [H]} \sum_{x^{-1}(x(\bar{j})+y)) \in \mathcal{N}(x^{-1}(x(\bar{\imath})+y))} \sigma_{x^{-1}(x(\bar{j})+y)}\big( \langle \varphi_{\text{qry}}^{(h)}(f(\bar{\imath}) + \rho(x^{-1}(x(\bar{\imath}) + y))),$$
$$\varphi_{\text{key}}^{(h)}(f(\bar{j}) + \rho(x^{-1}(x(\bar{j}) + y))) \rangle \big) \varphi_{\text{val}}^{(h)}(f(\bar{j})) \Big)$$

$$= \varphi_{\text{out}}\Big( \bigcup_{h \in [H]} \sum_{x^{-1}(x(\bar{j})+y)) \in \mathcal{N}(x^{-1}(x(\bar{\imath})+y))} \sigma_{x^{-1}(x(\bar{j})+y)}\big( \langle \varphi_{\text{qry}}^{(h)}(f(\bar{\imath}) + \rho^P(x(\bar{\imath}) + y)),$$
$$\varphi_{\text{key}}^{(h)}(f(\bar{j}) + \rho^P(x(\bar{j}) + y)) \rangle \big) \varphi_{\text{val}}^{(h)}(f(\bar{j})) \Big)$$

Here, we have used the substitution $\bar{i} = x^{-1}(x(i) - y) \Rightarrow i = x^{-1}(x(\bar{i}) + y)$ and $\bar{j} = x^{-1}(x(j) - y) \Rightarrow j = x^{-1}(x(\bar{j}) + y)$. Since the area of summation remains equal to any translation $y \in \mathbb{R}^d$, we have:

$$\sum_{x^{-1}(x(\bar{j})+y) \in \mathcal{N}(x^{-1}(x(\bar{i})+y))} [\cdot] = \sum_{x^{-1}(x(\bar{j})) \in \mathcal{N}(x^{-1}(x(\bar{i})))} [\cdot] = \sum_{\bar{j} \in \mathcal{N}(\bar{i})} [\cdot].$$

Hence, we can further reduce the expression above as:

$$m\big[\mathcal{L}_y[f], \rho\big](i)$$
$$= \varphi_{\text{out}}\Big( \bigcup_{h \in [H]} \sum_{\bar{j} \in \mathcal{N}(\bar{i})} \sigma_{\bar{j}}\big(\langle \varphi_{\text{qry}}^{(h)}(f(\bar{i}) + \rho^P(x(\bar{i}) + y)), \varphi_{\text{key}}^{(h)}(f(\bar{j}) + \rho^P(x(\bar{j}) + y))\rangle\big) \varphi_{\text{val}}^{(h)}(f(\bar{j})) \Big)$$

Since, $\rho(\bar{i}) + y \neq \rho(\bar{i})$ and $\rho(\bar{j}) + y \neq \rho(\bar{j})$, we are unable to reduce the expression further towards the form of $m(f, \rho)(i)$. Consequently, we conclude that the absolute positional encoding does not allow for translation equivariance either. $\qquad\square$

***Proof of Claim 4.3.*** If the self-attention formulation provided in Eq. 11 is translation equivariant, then it must hold that $m^r[\mathcal{L}_y[f], \rho](i) = \mathcal{L}_y[m^r[f, \rho]](i)$. Consider a translated input signal $\mathcal{L}_y[f](i) = f(x^{-1}(x(i) - y))$. The self-attention operation on $\mathcal{L}_y[f]$ is given by:

$$m^r\big[\mathcal{L}_y[f], \rho\big](i)$$
$$= \varphi_{\text{out}}\Big( \bigcup_{h \in [H]} \sum_{j \in \mathcal{N}(i)} \sigma_j\big(\langle \varphi_{\text{qry}}^{(h)}(\mathcal{L}_y[f](i)), \varphi_{\text{key}}^{(h)}(\mathcal{L}_y[f](i) + \rho(i, j))\rangle\big) \varphi_{\text{val}}^{(h)}(\mathcal{L}_y[f](j)) \Big)$$
$$= \varphi_{\text{out}}\Big( \bigcup_{h \in [H]} \sum_{j \in \mathcal{N}(i)} \sigma_j\big(\langle \varphi_{\text{qry}}^{(h)}(f(x^{-1}(x(i) - y))), \varphi_{\text{key}}^{(h)}(f(x^{-1}(x(j) - y))$$
$$+ \rho(i, j))\rangle\big) \varphi_{\text{val}}^{(h)}(f(x^{-1}(x(j) - y))) \Big)$$
$$= \varphi_{\text{out}}\Big( \bigcup_{h \in [H]} \sum_{x^{-1}(x(\bar{j})+y) \in \mathcal{N}(x^{-1}(x(\bar{i})+y))} \sigma_{x^{-1}(x(\bar{j})+y)}\big(\langle \varphi_{\text{qry}}^{(h)}(f(\bar{i})), \varphi_{\text{key}}^{(h)}(f(\bar{j})$$
$$+ \rho(x^{-1}(x(\bar{i}) + y), x^{-1}(x(\bar{j}) + y))\rangle\big) \varphi_{\text{val}}^{(h)}(f(\bar{j})) \Big)$$

Here, we have used the substitution $\bar{i} = x^{-1}(x(i) - y) \Rightarrow i = x^{-1}(x(\bar{i}) + y)$ and $\bar{j} = x^{-1}(x(j) - y) \Rightarrow j = x^{-1}(x(\bar{j}) + y)$. By using the definition of $\rho(i, j)$ we can further reduce the expression above as:

$$= \varphi_{\text{out}}\Big( \bigcup_{h \in [H]} \sum_{x^{-1}(x(\bar{j})+y) \in \mathcal{N}(x^{-1}(x(\bar{i})+y))} \sigma_{x^{-1}(x(\bar{j})+y)}\big(\langle \varphi_{\text{qry}}^{(h)}(f(\bar{i})), \varphi_{\text{key}}^{(h)}(f(\bar{j})$$
$$+ \rho^P(x(\bar{j}) + y - (x(\bar{i}) + y))\rangle\big) \varphi_{\text{val}}^{(h)}(f(\bar{j})) \Big)$$
$$= \varphi_{\text{out}}\Big( \bigcup_{h \in [H]} \sum_{x^{-1}(x(\bar{j})+y) \in \mathcal{N}(x^{-1}(x(\bar{i})+y))} \sigma_{x^{-1}(x(\bar{j})+y)}\big(\langle \varphi_{\text{qry}}^{(h)}(f(\bar{i})), \varphi_{\text{key}}^{(h)}(f(\bar{j}) + \rho^P(x(\bar{j}) - x(\bar{i})))\rangle\big) \varphi_{\text{val}}^{(h)}(f(\bar{j})) \Big)$$
$$= \varphi_{\text{out}}\Big( \bigcup_{h \in [H]} \sum_{x^{-1}(x(\bar{j})+y) \in \mathcal{N}(x^{-1}(x(\bar{i})+y))} \sigma_{x^{-1}(x(\bar{j})+y)}\big(\langle \varphi_{\text{qry}}^{(h)}(f(\bar{i})), \varphi_{\text{key}}^{(h)}(f(\bar{j}) + \rho(\bar{i}, \bar{j}))\rangle\big) \varphi_{\text{val}}^{(h)}(f(\bar{j})) \Big)$$

Since the area of the summation remains equal to any translation $y \in \mathbb{R}^d$, we have that:

$$\sum_{x^{-1}(x(\bar{j})+y) \in \mathcal{N}(x^{-1}(x(\bar{i})+y))} [\cdot] = \sum_{x^{-1}(x(\bar{j})) \in \mathcal{N}(x^{-1}(x(\bar{i})))} [\cdot] = \sum_{\bar{j} \in \mathcal{N}(\bar{i})} [\cdot].$$

Resultantly, we can further reduce the expression above as:

$$m^r\big[\mathcal{L}_y[f], \rho\big](i) = \varphi_{\text{out}}\Big( \bigcup_{h \in [H]} \sum_{\bar{j} \in \mathcal{N}(\bar{i})} \sigma_{\bar{j}}\big(\langle \varphi_{\text{qry}}^{(h)}(f(\bar{i})), \varphi_{\text{key}}^{(h)}(f(\bar{j}) + \rho(\bar{i}, \bar{j}))\rangle\big) \varphi_{\text{val}}^{(h)}(f(\bar{j})) \Big)$$
$$= m^r[f, \rho](\bar{i}) = m^r[f, \rho](x^{-1}(x(i) - y))$$
$$= \mathcal{L}_y\big[m^r[f, \rho]\big](i)$$

We see that indeed $m^r[\mathcal{L}_y[f], \rho](i) = \mathcal{L}_y[m^r[f, \rho]](i)$. Consequently, we conclude that the relative positional encoding allows for translation equivariance. We emphasize that this is a consequence of the fact that $\rho(x^{-1}(x(\bar{i}) + y), x^{-1}(x(\bar{j}) + y)) = \rho(\bar{i}, \bar{j}), \forall y \in \mathbb{R}^d$. In other words, it comes from the fact that relative positional encoding is invariant to the action of the translation group. $\qquad\square$

**Proof of Claim 5.1.** If the lifting self-attention formulation provided in Eq. 11 is $\mathcal{G}$-equivariant, then it must hold that $m_{\mathcal{G}\uparrow}^r[\mathcal{L}_g[f], \rho](i, h) = \mathcal{L}_g[m_{\mathcal{G}\uparrow}^r[f, \rho]](i, h)$. Consider a $g$-transformed input signal $\mathcal{L}_g[f](i) = \mathcal{L}_y \mathcal{L}_{\tilde{h}}[f](i) = f(x^{-1}(\tilde{h}^{-1}(x(i) - y)))$, $g = (y, \tilde{h})$, $y \in \mathbb{R}^d$, $\tilde{h} \in \mathcal{H}$. The lifting group self-attention operation on $\mathcal{L}_g[f]$ is given by:

$$m_{\mathcal{G}\uparrow}^r[\mathcal{L}_y \mathcal{L}_{\tilde{h}}[f], \rho](i, h)$$
$$= \varphi_{\text{out}}\Big( \bigcup_{h \in [H]} \sum_{j \in \mathcal{N}(i)} \sigma_j \big( \langle \varphi_{\text{qry}}^{(h)}(\mathcal{L}_y \mathcal{L}_{\tilde{h}}[f](i)), \varphi_{\text{key}}^{(h)}(\mathcal{L}_y \mathcal{L}_{\tilde{h}}[f](i) + \mathcal{L}_h[\rho](i,j)) \rangle \big) \varphi_{\text{val}}^{(h)}(\mathcal{L}_y \mathcal{L}_{\tilde{h}}[f](j)) \Big)$$

$$= \varphi_{\text{out}}\Big( \bigcup_{h \in [H]} \sum_{j \in \mathcal{N}(i)} \sigma_j \big( \langle \varphi_{\text{qry}}^{(h)}(f(x^{-1}(\tilde{h}^{-1}(x(i) - y)))), \varphi_{\text{key}}^{(h)}(f(x^{-1}(\tilde{h}^{-1}(x(j) - y))) + \mathcal{L}_h[\rho](i,j)) \rangle \big) \varphi_{\text{val}}^{(h)}(f(x^{-1}(\tilde{h}^{-1}(x(j) - y)))) \Big)$$

$$= \varphi_{\text{out}}\Big( \bigcup_{h \in [H]} \sum_{x^{-1}(\tilde{h}x(\bar{j})+y) \in \mathcal{N}(x^{-1}(\tilde{h}x(\bar{i})+y))} \sigma_{x^{-1}(\tilde{h}x(\bar{j})+y)} \big( \langle \varphi_{\text{qry}}^{(h)}(f(\bar{i})), \varphi_{\text{key}}^{(h)}(f(\bar{j}) + \mathcal{L}_h[\rho](x^{-1}(\tilde{h}x(\bar{i})+y), x^{-1}(\tilde{h}x(\bar{j})+y)) ) \rangle \big) \varphi_{\text{val}}^{(h)}(f(\bar{j})) \Big)$$

Here we have used the substitution $\bar{i} = x^{-1}(\tilde{h}^{-1}(x(i) - y)) \Rightarrow i = x^{-1}(\tilde{h}x(\bar{i}) + y)$ and $\bar{j} = x^{-1}(\tilde{h}^{-1}(x(j) - y)) \Rightarrow j = x^{-1}(\tilde{h}x(\bar{j}) + y)$. By using the definition of $\rho(i,j)$ we can further reduce the expression above as:

$$= \varphi_{\text{out}}\Big( \bigcup_{h \in [H]} \sum_{x^{-1}(\tilde{h}x(\bar{j})+y) \in \mathcal{N}(x^{-1}(\tilde{h}x(\bar{i})+y))} \sigma_{x^{-1}(\tilde{h}x(\bar{j})+y)} \big( \langle \varphi_{\text{qry}}^{(h)}(f(\bar{i})), \varphi_{\text{key}}^{(h)}(f(\bar{j}) + \rho^P(h^{-1}(\tilde{h}x(\bar{j})+y) - h^{-1}(\tilde{h}x(\bar{i})+y)) ) \rangle \big) \varphi_{\text{val}}^{(h)}(f(\bar{j})) \Big)$$

$$= \varphi_{\text{out}}\Big( \bigcup_{h \in [H]} \sum_{x^{-1}(\tilde{h}x(\bar{j})+y) \in \mathcal{N}(x^{-1}(\tilde{h}x(\bar{i})+y))} \sigma_{x^{-1}(\tilde{h}x(\bar{j})+y)} \big( \langle \varphi_{\text{qry}}^{(h)}(f(\bar{i})), \varphi_{\text{key}}^{(h)}(f(\bar{j}) + \rho^P(h^{-1}\tilde{h}(x(\bar{j}) - x(\bar{i}))) ) \rangle \big) \varphi_{\text{val}}^{(h)}(f(\bar{j})) \Big)$$

$$= \varphi_{\text{out}}\Big( \bigcup_{h \in [H]} \sum_{x^{-1}(\tilde{h}x(\bar{j})+y) \in \mathcal{N}(x^{-1}(\tilde{h}x(\bar{i})+y))} \sigma_{x^{-1}(\tilde{h}x(\bar{j})+y)} \big( \langle \varphi_{\text{qry}}^{(h)}(f(\bar{i})), \varphi_{\text{key}}^{(h)}(f(\bar{j}) + \mathcal{L}_{\tilde{h}^{-1}h}[\rho](\bar{i}, \bar{j})) \rangle \big) \varphi_{\text{val}}^{(h)}(f(\bar{j})) \Big)$$

Since, for unimodular groups, the area of summation remains equal for any $g \in \mathcal{G}$, we have that:

$$\sum_{x^{-1}(\tilde{h}x(\bar{j})+y) \in \mathcal{N}(x^{-1}(\tilde{h}x(\bar{i})+y))} [\cdot] = \sum_{x^{-1}(\tilde{h}x(\bar{j})) \in \mathcal{N}(x^{-1}(\tilde{h}x(\bar{i})))} [\cdot] = \sum_{x^{-1}(x(\bar{j})) \in \mathcal{N}(x^{-1}(x(\bar{i})))} [\cdot] = \sum_{\bar{j} \in \mathcal{N}(\bar{i})} [\cdot].$$

Resultantly, we can further reduce the expression above as:

$$m_{\mathcal{G}\uparrow}^r[\mathcal{L}_y \mathcal{L}_{\tilde{h}}[f], \rho](i, h)$$
$$= \varphi_{\text{out}}\Big( \bigcup_{h \in [H]} \sum_{\bar{j} \in \mathcal{N}(\bar{i})} \sigma_{\bar{j}} \big( \langle \varphi_{\text{qry}}^{(h)}(f(\bar{i})), \varphi_{\text{key}}^{(h)}(f(\bar{j}) + \mathcal{L}_{\tilde{h}^{-1}h}[\rho](\bar{i}, \bar{j})) \rangle \big) \varphi_{\text{val}}^{(h)}(f(\bar{j})) \Big)$$
$$= m_{\mathcal{G}\uparrow}^r[f, \rho](\bar{i}, \tilde{h}^{-1}h) = m_{\mathcal{G}\uparrow}^r[f, \rho](x^{-1}(\tilde{h}^{-1}(x(i) - y)), \tilde{h}^{-1}h)$$
$$= \mathcal{L}_y \mathcal{L}_{\tilde{h}}[m_{\mathcal{G}\uparrow}^r[f, \rho]](i, h).$$

We see indeed that $m_{\mathcal{G}\uparrow}^r[\mathcal{L}_y \mathcal{L}_{\tilde{h}}[f], \rho](i, h) = \mathcal{L}_y \mathcal{L}_{\tilde{h}}[m_{\mathcal{G}\uparrow}^r[f, \rho]](i, h)$. Consequently, we conclude that the lifting group self-attention operation is group equivariant. We emphasize once more that this is a consequence of the fact that $\mathcal{L}_g[\rho](i, j) = \rho(i, j)$, $\forall g \in \mathcal{G}$. In other words, it comes from the fact that the positional encoding used is invariant to the action of elements $g \in \mathcal{G}$. $\square$

**Proof of Claim 5.2.** If the group self-attention formulation provided in Eq. 11 is $\mathcal{G}$-equivariant, then it must hold that $m_{\mathcal{G}}^r[\mathcal{L}_g[f], \rho](i, h) = \mathcal{L}_g[m_{\mathcal{G}}^r[f, \rho]](i, h)$. Consider a $g$-transformed input signal $\mathcal{L}_g[f](i, \tilde{h}) = \mathcal{L}_y \mathcal{L}_{\bar{h}}[f](i, \tilde{h}) = f(\rho^{-1}(\bar{h}^{-1}(\rho(i) - y)), \bar{h}\tilde{h})$, $g = (y, \bar{h})$, $y \in \mathbb{R}^d$, $\bar{h} \in \mathcal{H}$. The group self-attention operation on $\mathcal{L}_g[f]$ is given by:

$$m_{\mathcal{G}}^r[\mathcal{L}_y \mathcal{L}_{\bar{h}}[f], \rho](i, h)$$
$$= \varphi_{\text{out}}\Big( \bigcup_{h \in [H]} \sum_{\hat{h} \in \mathcal{H}} \sum_{(j, \hat{h}) \in \mathcal{N}(i, \tilde{h})} \sigma_{j, \hat{h}} \big( \langle \varphi_{\text{qry}}^{(h)}(\mathcal{L}_y \mathcal{L}_{\bar{h}}[f](i, \tilde{h})), \varphi_{\text{key}}^{(h)}(\mathcal{L}_y \mathcal{L}_{\bar{h}}[f](j, \hat{h}) + \mathcal{L}_h[\rho]((i, \tilde{h}), (j, \hat{h}))) \rangle \big) \varphi_{\text{val}}^{(h)}(\mathcal{L}_y \mathcal{L}_{\bar{h}}[f](j, \hat{h})) \Big)$$

$$= \varphi_{\text{out}}\Big( \bigcup_{h \in [H]} \sum_{\hat{h} \in \mathcal{H}} \sum_{(j, \hat{h}) \in \mathcal{N}(i, \tilde{h})} \sigma_{j, \hat{h}} \big( \langle \varphi_{\text{qry}}^{(h)}(f(x^{-1}(\bar{h}^{-1}(x(i) - y)), \bar{h}^{-1}\tilde{h})), \varphi_{\text{key}}^{(h)}(f(x^{-1}(\bar{h}^{-1}(x(j) - y)), \bar{h}^{-1}\hat{h}) + \mathcal{L}_h[\rho]((i, \tilde{h}), (j, \hat{h}))) \rangle \big) \varphi_{\text{val}}^{(h)}(f(x^{-1}(\bar{h}^{-1}(x(j) - y)), \bar{h}^{-1}\hat{h})) \Big)$$

$$= \varphi_{\text{out}}\Big( \bigcup_{h\in[H]} \sum_{\bar{h}\tilde{h}'\in\mathcal{H}} \sum_{(x^{-1}(\bar{h}x(\bar{j})+y),\bar{h}\hat{h}')\in\mathcal{N}(x^{-1}(\bar{h}x(\bar{i})+y),\bar{h}\tilde{h}')} \sigma_{x^{-1}(\bar{h}x(\bar{j})+y),\bar{h}\hat{h}'}\big(\langle\varphi_{\text{qry}}^{(h)}(f(\bar{i},\tilde{h}')),\varphi_{\text{key}}^{(h)}(f(\bar{j},\hat{h}')+\mathcal{L}_{\bar{h}}[\rho]((x^{-1}(\bar{h}x(\bar{i})+y),\bar{h}\tilde{h}'),$$
$$(x^{-1}(\bar{h}x(\bar{j})+y),\bar{h}\hat{h}')))\rangle\varphi_{\text{val}}^{(h)}(f(\bar{j},\hat{h}'))\Big)$$

Here we have used the substitutions $\bar{i} = x^{-1}(\bar{h}^{-1}(x(i) - y)) \Rightarrow i = x^{-1}(\bar{h}x(\bar{i})+y))$, $\tilde{h}' = \bar{h}^{-1}\tilde{h}$, and $\bar{j} = x^{-1}(\bar{h}^{-1}(x(j) - y)) \Rightarrow i = x^{-1}(\bar{h}x(\bar{i})+y))$, $\hat{h}' = \bar{h}^{-1}\hat{h}$. By using the definition of $\rho((i,\tilde{h}),(j,\hat{h}))$ we can further reduce the expression above as:

$$= \varphi_{\text{out}}\Big( \bigcup_{h\in[H]} \sum_{\bar{h}\tilde{h}'\in\mathcal{H}} \sum_{(x^{-1}(\bar{h}x(\bar{j})+y),\bar{h}\hat{h}')\in\mathcal{N}(x^{-1}(\bar{h}x(\bar{i})+y),\bar{h}\tilde{h}')} \sigma_{x^{-1}(\bar{h}x(\bar{j})+y),\bar{h}\hat{h}'}\big(\langle\varphi_{\text{qry}}^{(h)}(f(\bar{i},\tilde{h}')),\varphi_{\text{key}}^{(h)}(f(\bar{j},\hat{h}')$$
$$+ \rho^P(\bar{h}^{-1}(\bar{h}x(\bar{j}) + y - (\bar{h}x(\bar{i})+y)),\bar{h}^{-1}\bar{h}\tilde{h}'^{-1}\hat{h}'))\rangle\varphi_{\text{val}}^{(h)}(f(\bar{j},\hat{h}'))\Big)$$

$$= \varphi_{\text{out}}\Big( \bigcup_{h\in[H]} \sum_{\bar{h}\tilde{h}'\in\mathcal{H}} \sum_{(x^{-1}(\bar{h}x(\bar{j})+y),\bar{h}\hat{h}')\in\mathcal{N}(x^{-1}(\bar{h}x(\bar{i})+y),\bar{h}\tilde{h}')} \sigma_{x^{-1}(\bar{h}x(\bar{j})+y),\bar{h}\hat{h}'}\big(\langle\varphi_{\text{qry}}^{(h)}(f(\bar{i},\tilde{h}')),\varphi_{\text{key}}^{(h)}(f(\bar{j},\hat{h}')$$
$$+ \rho^P(\bar{h}^{-1}\bar{h}(x(\bar{j}) - x(\bar{i}),\tilde{h}'^{-1}\hat{h}')))\rangle\varphi_{\text{val}}^{(h)}(f(\bar{j},\hat{h}'))\Big)$$

$$= \varphi_{\text{out}}\Big( \bigcup_{h\in[H]} \sum_{\bar{h}\tilde{h}'\in\mathcal{H}} \sum_{(x^{-1}(\bar{h}x(\bar{j})+y),\bar{h}\hat{h}')\in\mathcal{N}(x^{-1}(\bar{h}x(\bar{i})+y),\bar{h}\tilde{h}')} \sigma_{x^{-1}(\bar{h}x(\bar{j})+y),\bar{h}\hat{h}'}\big(\langle\varphi_{\text{qry}}^{(h)}(f(\bar{i},\tilde{h}')),\varphi_{\text{key}}^{(h)}(f(\bar{j},\hat{h}')$$
$$+ \mathcal{L}_{\bar{h}^{-1}h}[\rho]((\bar{i},\tilde{h}'),(\bar{j},\hat{h}')))\rangle\varphi_{\text{val}}^{(h)}(f(\bar{j},\hat{h}'))\Big)$$

Furthermore, since for unimodular groups the area of summation remains equal for any transformation $g \in \mathcal{G}$, we have that:

$$\sum_{(x^{-1}(\bar{h}x(\bar{j})+y),\bar{h}\hat{h}')\in\mathcal{N}(x^{-1}(\bar{h}x(\bar{i})+y),\bar{h}\tilde{h}')} [\cdot] = \sum_{(x^{-1}(\bar{h}x(\bar{j})),\bar{h}\hat{h}')\in\mathcal{N}(x^{-1}(\bar{h}x(\bar{i})),\bar{h}\tilde{h}')} [\cdot] = \sum_{(x^{-1}(x(\bar{j})),\hat{h}')\in\mathcal{N}(x^{-1}(x(\bar{i})),\tilde{h}')} [\cdot] = \sum_{(\bar{j},\hat{h}')\in\mathcal{N}(\bar{i},\tilde{h}')} [\cdot].$$

Additionally, we have that $\sum_{\bar{h}\tilde{h}'\in\mathcal{H}}[\cdot] = \sum_{\tilde{h}'\in\mathcal{H}}[\cdot]$. Resultantly, we can further reduce the expression above as:

$$m_{\mathcal{G}}^r[\mathcal{L}_y\mathcal{L}_{\bar{h}}[f],\rho](i,h) = \varphi_{\text{out}}\Big( \bigcup_{h\in[H]} \sum_{\tilde{h}'\in\mathcal{H}} \sum_{(\bar{j},\hat{h}')\in\mathcal{N}(\bar{i},\tilde{h}')} \sigma_{\bar{j},\hat{h}'}\big(\langle\varphi_{\text{qry}}^{(h)}(f(\bar{i},\tilde{h}')),\varphi_{\text{key}}^{(h)}(f(\bar{j},\hat{h}')+$$
$$\mathcal{L}_{\bar{h}^{-1}h}[\rho]((\bar{i},\tilde{h}'),(\bar{j},\hat{h}')))\rangle\varphi_{\text{val}}^{(h)}(f(\bar{j},\hat{h}'))\Big)$$
$$= m_{\mathcal{G}}^r[f,\rho](\bar{i},\bar{h}^{-1}h) = m_{\mathcal{G}}^r[f,\rho](x^{-1}(\bar{h}^{-1}(x(i) - y)),\bar{h}^{-1}h)$$
$$= \mathcal{L}_y\mathcal{L}_{\bar{h}}[m_{\mathcal{G}}^r[f,\rho]](i,h).$$

We see that indeed $m_{\mathcal{G}}^r[\mathcal{L}_y\mathcal{L}_{\bar{h}}[f],\rho](i,h) = \mathcal{L}_y\mathcal{L}_{\bar{h}}[m_{\mathcal{G}}^r[f,\rho]](i,h)$. Consequently, we conclude that the group self-attention operation on is group equivariant. We emphasize once more that this is a consequence of the fact that $\mathcal{L}_g[\rho]((i,\tilde{h}),(j,\hat{h})) = \rho((i,\tilde{h}),(j,\hat{h}))$, $\forall g \in \mathcal{G}$. In other words, it comes from the fact that the positional encoding used is invariant to the action of elements $g \in \mathcal{G}$. $\square$

