# OpenReview forum: "Group Equivariant Stand-Alone Self-Attention For Vision"
_ICLR.cc/2021/Conference — ICLR 2021 Poster_

### Official Review · AnonReviewer1 · 2020-10-28
**Initial review**

**Rating:** 6
**Confidence:** 5

**Review:**

***Summary***

I would firstly like to thank the authors for an interesting read. I enjoyed going through the submission very much.

The submission outlines a general framework to make self-attention equivariant under the action of ‘arbitrary’ groups. The authors go about laying out the mathematical foundations of self-attention from a functional standpoint and then find the conditions that must be satisfied by the individual components of the attention mechanism to make it adhere to the imposed equivariance constraints. Namely, they note that absolute positional encodings must be done away with and that activations must be lifted to a homogenous space of the group in question in order for non-trivial (non-constant) relative positional encodings to be used. They experiment on the MNIST-rot, CIFAR10 and PatchCameylon datasets, but unfortunately the results fall significantly behind standard baselines.



***Pros***

The paper is written very clearly and from what I could see the mathematics appears to be technically sound.

The authors also provide an in-depth appendix, which contains a lot of details as to group theory and their experiments.

I think the primary objective of making self-attention equivariant is nice. I also think that the conclusion the authors come to makes sense and in someways (with hindsight, having read the paper) is to be expected.

In the experiments, the choices of which datasets to use make sense from the point of view of the equivariance literature.

I like it how the authors explain that the networks are steerable by design because the group can act on the positional encodings.



***Cons and constructive feedback***

I would like to see a discussion comparing the proposed method with the works of “Affine Self-Convolution” by (Diaconu and Worrall, 2019), and “Attentive group equivariant convolutional networks” (Romero et al., 2019), which are both very close works in this area. What are the differences?

The results are quite disappointing. After the main theory I would have been happy with results en par with existing works, but CIFAR performance of, for instance, 83.7% compared to a baseline of 91.1% really isn’t great. For me this is the main drawback of this work. Furthermore, how are the models comparable with the baselines? Are they matched in terms of numbers of parameters, or size of the activations, or something else? Perhaps this could be a source of the discrepancy between the results.

I would also like to see comparisons with the work of Romero and Diaconu, who have results in their papers.

In the abstract you state that you can impost equivariance to arbitrary groups? Is this really so? Surely just compact, discrete groups?

Equation 1: you introduce the notation ‘sigma’ for the softmax nonlinearity but then immediately use \text{softmax} in eqn 1.

Equation 7: what is the difference between the right hand part of the first line and the second line?

Definition 4.1: is this definition not overly restrictive by focussing on left-regular representations for L and L’? It also precludes the use of irreducible representations as commonly used in the works on group equivariant networks.

Example 4.2.1 I think this could have been stated more simply by specifying the group and action.

Definition 4.3 shown -> show

Proposition 4..2 Nor permutation nor translation -> Neither permutation nor translation

Section 5.2: Good references here would be Theorem 3.1 of “A General Theory of Equivariant CNNs on Homogeneous Spaces”, (Cohen et al., 2018) and Theorem 1 of “B-Spline CNNs on Lie Groups” (Bekkers, 2020). Also, is this statement proven? I couldn’t find it in the submission.

Section 6. Efficient implementation…: I would like to see more detail here since it is not clear in which way the implementation is efficient.

Just a small note: you use American spelling throughout the paper, but missed out behaviour and neighbourhood, which should read behavior and neighborhood.

***Post-rebuttal review***

I have increasing my recommendation to a 6. This is on account of the improvements to the submission by the authors, a detailed rebuttal, and somewhat to align with the recommendations of the other reviewers. I still do find that the experimental results could be improved somewhat, but as another reviewer pointed out, this is not a huge concern within the majority of the equivariance literature. I thank the authors for an interesting read and thank you for a good rebuttal.

---

> ### Author Response · Authors · 2020-11-15
> **Initial Response R1**
>
> Dear reviewer 1,
>
> First of all we would like to thank you very much for thorough, insightful review. We are very happy to hear that you enjoyed reading the paper.
>
> Now we will answer to all of your questions, suggestions, comments and concerns:
>
> **I would like to see a discussion comparing the proposed method with the works of “Affine Self-Convolution” by (Diaconu and Worrall, 2019), and “Attentive group equivariant convolutional networks” (Romero et al., 2019), which are both very close works in this area. What are the differences?**
> We largely modified the related work section. We included additional references and better positioned our work w.r.t. related works, e.g., Romero et al., 2019, Romero et al., 2020a, Fuchs et al., 2020.
>
>
> **The results are quite disappointing. After the main theory I would have been happy with results en par with existing works, but CIFAR performance of, for instance, 83.7% compared to a baseline of 91.1% really isn’t great. For me this is the main drawback of this work. Furthermore, how are the models comparable with the baselines? Are they matched in terms of numbers of parameters, or size of the activations, or something else? Perhaps this could be a source of the discrepancy between the results.**
> We modified the first paragraph of the Experiments section to emphasize the relevance of our experimental results. Specifically, we emphasize that the focus of our experiments is the evaluation of GSA-Nets with respect to equivalent non-equivariant self-attention networks. Convolutional architectures are included in our results only to provide a fair view to the yet present gap between self-attention and convolutional architectures in vision tasks, which is also present for their group equivariant counterparts. In other words, our networks do not build upon these architectures. We note that this performance gap is not unique to our work but universal to self-attention. Self-attention still has problems in outperforming convolutional nets for visual data. In fact this is a very active research direction, for which alternatives are also submitted to this ICLR 2021 (with good assessment scores) https://openreview.net/forum?id=YicbFdNTTy. However, shall self-attention networks outperform convolutional networks, we expect that by making them group equivariant additional improvements can be obtained.
>
> **I would also like to see comparisons with the work of Romero and Diaconu, who have results in their papers.**
> We included results from attentive G-CNNs (Romero et. al. 2020a) in our tables. We do not add results from Diaconu & Worrall as they use different datasets or groups in their experiments.
>
>
> **In the abstract you state that you can impost equivariance to arbitrary groups? Is this really so? Surely just compact, discrete groups?**
> We note that our self-attention *formulation*  is valid for continuous groups as well (for theoretical purposes). Though it is true that networks using this formulation cannot directly handle continuous groups in practice, we see that this is not detrimental in practice.  We have explicitly stated this behavior in a new paragraph above Section 5.1.
> Specifically, using fine enough discrete groups seems to be sufficient.  This statement is based upon empirical evidence from our work as well as from extensive experiments in Weiler & Cesa, 2019, Tab 3, which indicate that performance saturates for fine discrete approximations of the underlying continuous group. In addition, networks handling fine enough discrete approximations consistently outperform networks handling continuous groups via irreducible representations. We further conjecture that this is a result of the networks receiving discrete signals as input. As the action of several group elements fall within the same pixel, no further improvement can be obtained.
>
> **Equation 1: you introduce the notation ‘sigma’ for the softmax nonlinearity but then immediately use \text{softmax} in eqn 1.**
> This has been resolved in the new version of the manuscript.
>
> **Equation 7: what is the difference between the right hand part of the first line and the second line?** This was a typo. This has been resolved in the new version of the manuscript.

---

> > ### Author Response · Authors · 2020-11-15
> > **Initial Response R1 -- Continuation --**
> >
> > **Definition 4.1: is this definition not overly restrictive by focussing on left-regular representations for L and L’? It also precludes the use of irreducible representations as commonly used in the works on group equivariant networks.**
> > This can indeed be extended to other type of representations. However, we restrict ourselves to regular representations as we exclusively develop theory and experiments for that kind of representations. Adding more general representations could elevate the difficulty of the paper and occlude its contributions. In addition, this helps us to position our work better w.r.t. existing literature (e.g., SE(3)-Transformers, Fuchs et. al., 2020).
> > We note that the selection of regular representations is not arbitrary but based on theoretical and experimental indicating that they are more expressive than the irreducible ones (Weiler & Cesa, 2019; Ravanbakhsh, 2020).
> >
> > **Example 4.2.1 I think this could have been stated more simply by specifying the group and action.**
> > We note that we construct this example using the action (or representation of the group) for set elements. Note that set elements the set $i \in \mathcal{S} $ cannot be translated as translation is not well-defined in the space of set indexes. In order to do so, the set element must be taken to an space where this action is well-defined ($\mathcal{X}$ in the example). This is performed via the position mapping $x: i \mapsto x(i) \in \mathcal{X}$. Once in this space, we can now apply a translation to the position of the index. For a translation given by $y$, its action is given by $x'(i) = x(i) + y$. However, as the function $f$ in Ex. 4.2.1 is defined on the set indexes space, we need to go back to this space. Consequently, we have that  $i' = x^{-1}(x(i) + y)$.
> >
> > As a result, we see that the equivalent of the action of the group for the set space is given by $y * i = x^{-1}(x(i) + y)$ and the group representation for functions defined in this set is given by $\mathcal{L}_{y}[f] (i) = f(y^{-1}*i) = f(x^{-1}(x(i) - y))$. This gives us the description shown in Example 4.2.1.  For more details on this operation please see Appx. D.
> >
> > **Definition 4.3 shown -> show**
> > This has been resolved in the new version of the manuscript.
> >
> > **Proposition 4..2 Nor permutation nor translation -> Neither permutation nor translation**
> > This has been resolved in the new version of the manuscript.
> >
> > **Section 5.2: Good references here would be Theorem 3.1 of “A General Theory of Equivariant CNNs on Homogeneous Spaces”, (Cohen et al., 2018) and Theorem 1 of “B-Spline CNNs on Lie Groups” (Bekkers, 2020).**
> > We have included these references in the new version of the manuscript.
> >
> > **Also, is this statement proven? I couldn’t find it in the submission.**
> > It is not explicitly proven in the document. However, it is straightforwardly derived from the proof of equivariance for group self-attention (Appx. G).  In order to see this, it is sufficient to put away the non-linearities in the operation for a moment. I.e., define a "linear self-attention". From here one can conclude that the operation is equivariant as well (while being defined on set elements).  From here we can conclude the validity of the statement. We considered this to be sufficient for the submission. However, if you would like to see this explicitly in the document, we are happy to include this specifically in Appx. G.
> >
> > **Section 6. Efficient implementation…: I would like to see more detail here since it is not clear in which way the implementation is efficient.** It is efficient in the sense that a naive implementation would require recomputing content attention values for each $\mathcal{L}_{h}[\rho], h \in \mathcal{H}$.
> >
> > **Just a small note: you use American spelling throughout the paper, but missed out behaviour and neighbourhood, which should read behavior and neighborhood.**
> > This has been resolved in the new version of the manuscript. Thank you for pointing this out.
> >
> > Once again, thank you very much for your time, attention and very useful commentaries. We sincerely appreciate the time you put in evaluating our approach. If you have any further questions or comments please let us know. We are more than happy to answer them :)
> >
> > Best regards,
> > The Authors.

---

### Official Review · AnonReviewer4 · 2020-10-28
**An interesting paper with good evaluation**

**Rating:** 8
**Confidence:** 5

**Review:**

### Summary
The paper proposes an approach for building group-equivariant self-attention networks. The authors use group-invariant positional encodings, and thus the output of a self-attention layer is equivariant under the actions from the considered group. They empirically demonstrate that group-equivariant self-attention networks have an improvement over their non-equivariant counterparts.

### Pros
1. I enjoyed reading the paper. It has a clear title. The abstract, the intro, and the related work are well-written and give a clear understanding of the content of the paper. The technical sections are coherent and well-structured.
2. The authors propose a general formulation of group-equivariant self-attention networks and rigorously prove their statements.
3. The experiments demonstrate the improvements achieved by the added group-equivariance in image classification on several datasets.

### Cons
I did not find any major weaknesses in the presented paper. I did find some minor issues which, however, do not change the general impression and my decision.
1. While the authors claim that the proposed formulation is general, they only consider 2 compact groups. A group of reflections and a discrete group of rotations. The authors also claim that current computational constraints restrict the possibility of training models with big vicinities until they converge. This issue can be especially important if a group of dilations (scaling group) is considered. The current scale-equivariant models of Bekkers 2020, Sosnovik et al. 2020, Worrall and Welling 2019, Romero et al. 2020 require one to process images with big filters. Is an implementation of a scale-equivariant self-attention network feasible given the current constraints? A discussion of these limitations would make the contribution of the current paper more clear.
2. The authors use long equations with a very nested structure. Reading these equations is complicated given the current notation. The main source of confusion comes from repetitive sequences of brackets. A minor rethinking of the notation may improve the readability. For example, using square brackets when a function is an argument i.e. $\alpha(f)(i, j) \rightarrow \alpha[f](i, j)$

It is a well-written paper. The theory is coherent. The text is easy-to-follow. The experimental section demonstrates a thorough evaluation of the proposed method.

### After-rebuttal comments
The authors andwered my questions. My decision stays the same

---

> ### Author Response · Authors · 2020-11-15
> **Initial Response R4**
>
> Dear reviewer 4,
>
> First of all we would like to thank you very much for thorough, insightful review. We are very happy to hear that you enjoyed reading the paper and happy to see that you strongly support our work. Thank you!
>
> In the following we elaborate upon the minor issues you raised:
>
> **The authors also claim that current computational constraints restrict the possibility of training models with big vicinities until they converge. This issue can be especially important if a group of dilations (scaling group) is considered. The current scale-equivariant models of Bekkers 2020, Sosnovik et al. 2020, Worrall and Welling 2019, Romero et al. 2020 require one to process images with big filters. Is an implementation of a scale-equivariant self-attention network feasible given the current constraints? A discussion of these limitations would make the contribution of the current paper more clear.**
> This is a very important point indeed. In fact is difficult to come up with a working scale-equivariant self-attention network given the current constraints. Luckily, there is an option. It is possible to replicate the technique used in Worrall & Welling, 2019, for self-attention. Worrall & Welling, 2019 implement their method in terms of dilated convolutions by using a dyadic set of scales (see Fig. 3 of their paper). We believe that an equivalent self-attentional counterpart can be constructed. To this end, one can consider a "sparse" dilation of the neighborhood in self-attention in a way similar to dilated convolutions. This is an interesting application. However, the question remaining is how good would such a network be. Worrall & Welling, 2019 reported that this did not work very well in practice. We will briefly discuss this in more detail in Appx. E.
>
> **The authors use long equations with a very nested structure. Reading these equations is complicated given the current notation. The main source of confusion comes from repetitive sequences of brackets. A minor rethinking of the notation may improve the readability. For example, using square brackets when a function is an argument**
> We implemented your bracket's suggestion across the paper. Indeed the readability remarkably improved. We also improve Eqs. 13 and 14 by (1) describing lifting and group self-attention in terms of vanilla self-attention, and (2) providing an intuitive description of their modus operandi.
>
> Once again, thank you very much for your time, attention and very useful commentaries. We sincerely appreciate the time you put in evaluating our approach. If you have any further questions or comments please let us know. We are more than happy to answer them :)
>
> Best regards,
> The Authors

---

### Official Review · AnonReviewer5 · 2020-11-06
**good progress, exposition and paper balance not sufficiently good....**

**Rating:** 6
**Confidence:** 3

**Review:**

This paper develops self-attention mechanisms for group equivariant networks. This is a natural and potentially useful piece of architecture, that, drawing inspiration from its success for text and image analysis, is likely to introduce an effective inductive bias to the group convolution networks.
The paper is based on the following series of observations. First, represent an image as a set of $N$ feature vectors (e.g.,  pixels). Second, consider group actions $\mathcal{G}$ that can be represented as sub-groups of the permutation group $ֿ\mathbb{S}_N$ acting on the $N$ image pixels. Third, the standard self-attention layer is permutation equivariant. A way to further restrict this equivariance to smaller sub-group of the permutations  $\mathcal{G}<\mathbb{S}_N$ is by adding (or concatenating?) to the set of feature vectors, used as keys in the attention layer, functions which are invariant to this subgroup but not to any larger group. This will lead to operators that are equivariant to the desired sub-group $\mathcal{G}$. Lastly, since doing that directly on the representation of an image leads to larger invariances than desired, the authors work with signals defined over the group and consider regular group action (i.e., change of variables of functions). For example, for the roto-translation group that adds $90$ degrees rotations around the center of the image to the standard translations, this means representing four copies of the image and defining the self-attention layer from the original image and such signals as well as self-attention maps between pairs of such signals.

Overall, I feel this work offers some interesting observations relating symmetry and equivariance/invariance and self-attention. However, I find this work not fully ready in terms of exposition, ideas delivery and elaboration. The paper is not well balanced between developing and explaining its main contribution (starting page 6), and self-attention background. I understand there is much to cover, but explaining the main constructions in half a page is too short. I think the paper keeps reintroducing positional encoding and self-attention in several formulations and there is much space to save there. Furthermore, the explanations and notations are somewhat not coherent and clear, for example (eqs. 13, 14) are not sufficiently explained and demonstrated. Although i dont think it is a necessary condition for publication of such paper, the results don't show an improvement over "standard" group convolution methods, although the authors claim expressiveness of their self-attention networks is at-least of that of generalized convolutions. Lastly, the last paragraph before section 6, describing the expressive power of these network is not sufficiently developed and justified.

Some more comments are:
- There are different notations for concat (word and union) and softmax (word and sigma).
- Equation 7 has some repeating unclear part.
- $X_i$ is not defined in equation 5.
- What happens when the group action cannot be represented as subgroup of permutations? (Page 4, lower part). That is, when there is no such injective map as $x$...
- What is $m$ in equation in Proposition 4.1?
- Can $\mathcal{X}$ be continuous space or should is be discrete as well? If continuous how does $y\in\mathcal{X}$ in example 4.2.1 work?
- Sentence after proposition 4.2 is not clear.
- After definition 4.3 - "we shown" - where?
- What is $m^r$?
- "This allows us to go beyond group discretizations that live in the grid without introducing interpolation artifacts." --> How?
- Before section 5.1 - what is $\mathcal{H}$?
- In Proposition 5.2: shouldn't $\mathcal{L}_g$ also be applied to $\rho$?
- Page 6: "an space".
- Page 7: what are linear mappings on $\mathcal{G}$?
- Last paragraph before section 6: why is this layer maximally expressive for *sub-groups* of $\mathbb{S}_N$.

UPDATE:
I thank the authors for their detailed response and edits. Overall, the authors have addressed my comments, but I have failed to understand some parts. I am still (slightly) positive about this work. >> Please check again for typos.

---

> ### Author Response · Authors · 2020-11-15
> **Initial Response R5**
>
> Dear reviewer 5,
>
> First of all we would like to thank you very much for thorough, insightful review and for supporting our work.
>
> Firstly, we would like to clarify an important point of our work:
> We note that our group self-attention is not only restricted to sub-groups of the permutation group. For example, in our experiments we consider multiple groups of rotations finer than 90 degrees, which cannot be described in terms of permutations and thus, are not a subgroup of  $\mathbb{S}_{N}$. This comes from the fact that the positional encoding is defined in a continuous space, which is not restricted by any discrete grid (See Sec.5.1 for further details). **[EDIT]** We also modified the sentence after Def. 4.3 to accentuate that our approach is not limited to sub-groups of the permutation group.
>
> Based on this paragraph we now respond to the following comments:
>
> **What happens when the group action cannot be represented as a subgroup of permutations? (Page 4, lower part). That is, when there is no such injective map as $x$...**
> The map $x$ is still injective for groups not contained in the permutation group (See example below and Sec.5.1).
>
> **Can $\mathcal{X}$  be continuous space or should is be discrete as well? If continuous how does $y \in \mathcal{X}$ in example 4.2.1 work?**
> Yes, $\mathcal{X}$ is in fact continuous. Let us illustrate this with an example. Imagine that we take a picture of a flying bird with a clear sky as background. Then we represent the picture by a set $S=${$ i $}$_{i \in S}$ . In this case, the position function $x(i)$ is given by the corresponding spatial coordinates $x(i) \forall i \in S$. Now, imagine that the bird flies $y$ centimeters further. We take our camera and get another picture.  Once more, we represent the picture by a set $S=${$ i' $}$ _{i' \in S}$ with corresponding spatial coordinates $x(i') \forall i' \in S$. Note that the sets are equal as the resolution of the camera does not change. Also the positional encoding is equivalent as the space where the coordinates are defined ($\mathcal{X}$) does not change. We can see that the pictures are exactly equal except for the values of the position function, which are now given by  $x(i') = x(i) + y$. As a result, we see that this operation is equivariant to translations. In particular the "pixel" $i'$ is equivalent to the "pixel" $i$ up to a transformation. Specifically we have that $x'(i') = x(i) + y$ or equally $i' = x'^{-1}(x(i) + y)$. The same logic holds for other transformations such as fine-grained rotations.
>
> Now we proceed with answering the rest of your questions, suggestions, comments and concerns:
>
> **A way to further restrict this equivariance to smaller sub-group of the permutations $\mathcal{G}<\mathbb{S}_N$ is by adding (or concatenating?) to the set of feature vectors, used as keys in the attention layer, functions which are invariant to this subgroup but not to any larger group.**
> This is indeed the case. In order to demonstrate this, it is sufficient to follow the same steps as in the proof of the summation case considered in this work. The proofs boil down to a change of variables, which can also be done in the case of concatenations.
>
>  **Overall, I feel this work offers some interesting observations relating symmetry and equivariance/invariance and self-attention. However, I find this work not fully ready in terms of exposition, ideas delivery and elaboration. The paper is not well balanced between developing and explaining its main contribution (starting page 6), and self-attention background. I understand there is much to cover, but explaining the main constructions in half a page is too short**
> We understand that half a page to explain the main contribution is indeed too short. We have extended it in the new version of our manuscript and elaborated on some important properties of our approach (e.g., steerability). However, we respectfully disagree with your statement regarding the balance between the development and explanation of the paper's main contribution. An alternative to the current paper structure would be to instantiate right away Sec. 5 after introducing self-attention in Sec 3 and proceed with a "top-down" disentanglement of ideas. We considered this structure while constructing the paper but concluded that that structure made the paper much more difficult to read. As stated by some of the other reviewers, the paper is "easy-to-follow" and "makes sense with hindsight having read the paper". We argue that this is due to the progressive construction of ideas in Sec. 4, which is why we consider placing Sec 4 vital in order to progressively understand the contributions of the paper. In addition, we note that the contributions of the paper already start in Sec 4. In fact, to the best of our knowledge, several of the conclusions drawn in these part have not been proven before: Prop. 4.2, 4.3 and Sec. 4.3. We hope this sheds light into the importance of our structure selection.

---

> > ### Author Response · Authors · 2020-11-15
> > **Initial Response R5 -- Continuation --**
> >
> > **Furthermore, the explanations and notations are somewhat not coherent and clear, for example (eqs. 13, 14) are not sufficiently explained and demonstrated.**
> > We filtered out the incoherences mentioned by the reviewers in the new version of our work. In addition, We added a line intuitively explaining the definition of lifting and group self-attention right before the corresponding mathematical definitions. Additionally, we added a line in both definitions defining lifting and group self-attention in terms of vanilla self-attention (Eqs. 13, 15). We hope this makes them more accessible and gives more clarity on their modus operandi.
> >
> > **Although i dont think it is a necessary condition for publication of such paper, the results don't show an improvement over "standard" group convolution methods, although the authors claim expressiveness of their self-attention networks is at-least of that of generalized convolutions.**
> > This is true. However, we note that this is not the focus of the experimental part. Our focus is rather to evaluate GSA-Nets w.r.t. equivalent non-equivariant counterparts (the G-CNNs are included only for fair comparison) . Please see "Summary of changes in the new version of our work." for a broader discussion on our experimental results.
> >
> > **Lastly, the last paragraph before section 6, describing the expressive power of these network is not sufficiently developed and justified.**
> > We will add a detailed discussion on this behavior in a new section of the appendix, which will be linked to the corresponding paragraph.
> >
> > **There are different notations for concat (word and union) and softmax (word and sigma).**
> > This has been resolved in the new version of the manuscript.
> >
> > **Equation 7 has some repeating unclear part.**
> > This has been resolved in the new version of the manuscript.
> >
> > **$X_i$ is not defined in equation 5.** This has been resolved in the new version of the manuscript. We now define $i, j$ before Eq. 5.
> >
> > **What is $m$ in equation in Proposition 4.1?**
> > $m$ stands for multi-head self-attention. Please see Eqs. 9, 10.
> >
> > **Sentence after proposition 4.2 is not clear.**
> > We have rewritten this sentence.
> >
> > **After definition 4.3 - "we shown" - where?**
> > We built a semi-formal construction of why this is the case in Sec. 4.3. and Sec. 5. The formal proofs of our statements are given in Appx. G.
> >
> > **What is $m^{r}$ ?**
> > $m^{r}$ stands for multi-head self-attention with relative positional encodings. Please see Eq. 11.
> >
> > **"This allows us to go beyond group discretizations that live in the grid without introducing interpolation artifacts." --> How?**
> > This is due to the nature of the positional encoding, which is defined in a continuous space. Please see Sec. 5.1. in the new version of our paper.
> >
> > **Before section 5.1 - what is \mathcal{H}?**
> > $\mathcal{H}$ comes from the definition of an affine group $\mathcal{G} = \mathbb{R}^{d} \rtimes \mathcal{H}$, where $\mathcal{H}$ is a group acting on $\mathbb{R}^{d}$. Please see Def. C.3. for further details.
> >
> > **In Proposition 5.2: shouldn't  $\mathcal{L}_{g}$ also be applied to $\rho$ ?**
> > No. The reason is that the positional encoding is a construction of the operation and does not change as a function of the input. From page 4 in the absolute position paragraph:  "Note that this encoding is not dependent on functions defined on the set but only on the set itself. footnote - Illustratively, one can think of this as a function returning a vector representation of pixel positions in a grid. Regardless of any transformation performed to the image, the labeling of the grid itself remains exactly equal. -"
> >
> > **Page 6: "an space".**
> > This has been resolved in the new version of the manuscript.
> >
> > **Page 7: what are linear mappings on $\mathcal{G}$?**
> > A linear mapping on $\mathcal{G}$ is a linear map whose domain is not $\mathbb{R}^{d}$ anymore but the group itself $\mathcal{G}$.
> >
> > **Last paragraph before section 6: why is this layer maximally expressive for sub-groups of $\mathbb{S}_N$.**
> > This statement holds not only for sub-groups of $\mathbb{S}_N$. Ravanbaksh (2020) recently proved that networks that build upon layers using regular representations are equivariant universal approximators provided global receptive fields. Since global group self-attention has a global receptive field and uses regular representations, it fulfills the requirements given by  Ravanbaksh (2020)'s theory. **[EDIT]:** We have now extended and developed the last paragraph of this section.
> >
> > Once again, thank you very much for your time, attention and very useful commentaries. We sincerely appreciate the time you put in evaluating our approach. If you have any further questions or comments please let us know. We are more than happy to answer them :)
> >
> > Best regards,
> > The Authors.

---

### Official Review · AnonReviewer3 · 2020-11-09
**This paper studies group equivariance properties of self-attention networks. Permutation equivariance follows from the self-attention definition while group equivariance depends on the definition of the positional encoding.**

**Rating:** 7
**Confidence:** 4

**Review:**

The authors describe their contributions in the introduction: the analysis of equivariance of self-attention, and how group invariance in the relative positional encoding enables group equivariance of the self-attention.


The authors have a very condensed related work section without going into any detail but with a lot of citations on (non attention) papers about equivariance. Works missed include Equivariant transformer networks by Tai et al., Equivariant multi-view networks (Esteves et al.), SO(3)-equivariant representations (Esteves et al. ), and early work on equivariant function spaces by Hel-Or and Teo (Canonical Decomposition of Steerable Functions).

But probably most relevant is the missing discussion on equivariance on set networks (incl point cloud networks) and graph networks (Maron and Lipman). There is no positional encoding in point clouds but the value at each element are the coordinates and some parallels can be drawn to positional encodings (for example first few layers of pointnet).

The authors clearly define self-attention, first with matrices, and then with a functional formulation. The functional formulation is elegant but difficult to follow. The concatenation in the multi-head case is an example of where the vector space formulation allows replacing concatenation with a union. The authors might want to explain the benefits compared to tensor formulation.

Authors first prove permutation equivariance of global self-attention without positional encoding.
The proof in G.4.1 is kind of convoluted and might be clearer using only matrices (if $\Pi$ is permutation than $\sigma(\Pi Q K^T \Pi^T)\Pi V = \Pi \sigma(Q K^T) V$.

Then they prove translational equivariance for relative positional encoding. It is easy to see that relative positional encoding is translation invariant. The step to equivariance has to be followed in the appendix and it would be easier for the reader to provide at least a sketch in the main text.

The observation about translation paves the ground for generalizing to any group if the positional encoding is invariant to this group. Unfortunately, at this point the discussion becomes very confusing compared to the original "lifting" by Cohen and Welling (2016). While this paper makes the lifting appear as a trick, the main idea of Cohen and Welling is that when one applies group convolution on a quotient space, for example SE(2)-convolution on R^2, the result is automatically invariant in SO(2) since R^2=SE(2)/SO(2). Instead, one performs a group correlation where the output is a function of the group action (not of the quotient space) and after this step one applies a group convolution (also appearing in the spherical CNNs (Cohen)as well as in the icosahedral multi-view networks (Esteves)).

The paper concludes with the claim that linear mappings whose positional encoding is G-invariant are G-equivariant. This is easy to see in the definition of convolution and one can imagine this for linear mappings but it is difficult to see that self-attention is a linear mapping if one looks at eq. 5 and possible definitions of the encoding function (12). I see it through the equivalence proof in Cordonnier but not through the self-attention definition.

Experiments do not show any advantage of equivariant self-attention in z2 or r4 CNNS.

To summarize the paper is interesting but quite difficult to follow. I wish the paper would follow the tensor notation like in the SE(3)-transformers. Authors should justify the superiority of their formalism vs tensors.

Related work should not be condensed with mere listing of citations like \cite{*).

Last: Steerability is claimed in the abstract and the introduction but never mentioned again in the paper. One can somehow see how the positional encoding implies it but a section would be worth to be dedicated to it.

---

> ### Author Response · Authors · 2020-11-15
> **Initial Response R3**
>
> Dear reviewer 3,
>
> First of all we would like to thank you very much for thorough, insightful review and the time you invested in evaluating our work. We would also like to thank you for supporting our work.
>
> Now we will answer to all of your questions, suggestions, comments and concerns:
>
> **The authors have a very condensed related work section without going into any detail ....**
> We have largely modified the related work section. We incorporated several of the references you specified and positioned our work better w.r.t. related works.
>
>
> **But probably most relevant is the missing discussion on equivariance on set networks (incl point cloud networks) and graph networks (Maron and Lipman).**
> Due to space constraints we unfortunately were not able to include a discussion in comparison to set networks. We note, however, it is not standard to find set-networks in group equivariant literature as set equivariant architectures manage equivariance in a different way. With that being said, we did include the work of Maron et al 2020 "On Learning Sets of Symmetric Elements". This work bridges these two families of models and we believe our insights nicely connect with their work and they might be useful for works in that direction.
>
> **There is no positional encoding in point clouds but the value at each element are the coordinates and some parallels can be drawn to positional encodings (for example first few layers of pointnet).**
> We have briefly discussed the possibilities to use our work for graphs and other set-like structures as possible future direction by highlighting the connection between positional encodings and spatial coordinates (Sec. 7).
>
> **The functional formulation is elegant but difficult to follow. The authors might want to explain the benefits compared to tensor formulation.**
> We agree that for permutations a formulation using matrices can be more appropriate. However, in the general spectrum of our work, matrix notation can be much more cumbersome than functional notation. For example, in order to represent translations by $t = (t_x, t_y)$, an important running example in our work, one must define a matrix [[0,0, $t_x$],[0,0,$t_y$]] and append a row of ones to the coordinates of the object being translated. Next, this procedure needs to be repeated to each of the coordinates of the object to be translated.
>
> This procedure can quickly become cumbersome and difficult to understand, especially when these matrices change for different groups. This is for example the case for permutations w.r.t. the other groups considered in this work. Furthermore, matrix notation by itself can be difficult to understand for readers not acquainted with it. In order to avoid this, we decided to define the transformations in a way much more familiar to all of us which we have been using since high-school: a functional representation. For example, the translation of an object $f$ becomes $f(p_x-t_x, p_y-t_y)$ or simply $f(p - t)$. We note that the SE(3)-Transformer paper does not use tensor notation to define the convolution or prove translation equivariance.
>
> **Then they prove translational equivariance for relative positional encoding. It is easy to see that relative positional encoding is translation invariant. The step to equivariance has to be followed in the appendix and it would be easier for the reader to provide at least a sketch in the main text.**
> This is indeed a valid point. Unfortunately, we did not have space to incorporate this in the main text. However, as you state, "it is easy to see that relative positional encoding is translation invariant". Considering the current constraints as well as the comments from the other reviewers, we decided to leave this part as it is. We hope you understand our decision.
>
> **The observation about translation paves the ground for generalizing to any group if the positional encoding is invariant to this group. Unfortunately, at this point the discussion becomes very confusing compared to the original "lifting" by Cohen and Welling (2016). While this paper makes the lifting appear as a trick, the main idea of Cohen and Welling...**
> We apologize for the confusion in this paragraph. We have modified this paragraph to make it clearer. We now provide an analog to your sentence "...one performs a group correlation where the output is a function of the group action..." for the self-attention operation.

---

> > ### Author Response · Authors · 2020-11-15
> > **Initial Response R3 -- Continuation --**
> >
> > **The paper concludes with the claim that linear mappings whose positional encoding is G-invariant are G-equivariant. This is easy to see in the definition of convolution and one can imagine this for linear mappings but it is difficult to see that self-attention is a linear mapping if one looks at eq. 5 and possible definitions of the encoding function (12). I see it through the equivalence proof in Cordonnier but not through the self-attention definition.**
> > In order to see this, it is sufficient to put away the non-linearities in the operation  for a moment. I.e., define a "linear self-attention". After one concludes that the operation is equivariant, one can proceed to analyze if the non-linearities disturb the equivariance property. This is inline with conventional procedures when analyzing the equivariance properties of convolutional architectures. Initially one considers a convolution without non-linearity and subsequently demonstrates that the non-linearity does not disrupt equivariance, which happens to be the case for the softmax operation.
> >
> > **Experiments do not show any advantage of equivariant self-attention in z2 or r4 CNNS.**
> > This is true. However, we note that this is not the focus of the experimental part. Our focus is rather to evaluate GSA-Nets w.r.t. equivalent non-equivariant counterparts. Please see "Summary of changes in the new version of our work." for a broader discussion on our experimental results.
> >
> > **Last: Steerability is claimed in the abstract and the introduction but never mentioned again in the paper. One can somehow see how the positional encoding implies it but a section would be worth to be dedicated to it.**
> > This is an important point. We have included a section on this topic **Sec 5.1. Group self-attention is an steerable operation**.
> >
> > Once again, thank you very much for your time, attention and very useful commentaries. We sincerely appreciate the time you put in evaluating our approach. If you have any further questions or comments please let us know. We are more than happy to answer them :)
> >
> > Best regards,
> > The Authors.

---

### Author Response · Authors · 2020-11-15
**Summary of changes in the new version of our work.**

Dear reviewers,

First of all we would like to thank you very much for your thorough and valuable reviews and for the time you invested in writing them.  We have submitted a new revision of our work based on your observations. The quality of our manuscript has much improved based on your suggestions and we sincerely thank you for that.

In this comment we briefly summarize general changes to the document.  We will promptly address each of your individual concerns and questions as responses to each of your reviews.

#### Changes
**General**:
- We have improved the notation throughout the document (**R4**) and filtered out spelling inconsistencies and typos (**R1, R5**)

**Sec. 2. Related Work**:
- We largely modified the related work section. We included additional references suggested by the reviewers and better positioned our work w.r.t. related works (**R1, R3**).

**Sec. 5. Group Equivariant Stand-Alone Self-Attention**

* We have included an additional subsection **Sec. 5.1. Group self-attention is an steerable operation**. In this section we illustrate that (1) self-attention is a steerable operation (**R3**), and that (2) group self-attention is able to handle groups not contained in $\mathbb{S}_{N}$ (**R5**).

* We have added a line intuitively explaining the definition of lifting and group self-attention before the corresponding mathematical definitions. In addition, we added a line in both definitions that define lifting and group self-attention in terms of vanilla self-attention (**R3, R4, R5**).

* We have now further extended and developed the discussion of the last paragraph of Sec. 5.3 on the expressivity of group self-attention (**R5**).

**Sec. 6. Experiments**
* We modified the first paragraph to emphasize the relevance of our experimental results (**R1, R3, R5**).
Specifically, we emphasize that the focus of our experiments is the evaluation of GSA-Nets with respect to *equivalent* non-equivariant self-attention networks. Convolutional architectures are included in our results *only* to provide a fair view to the yet present gap between self-attention and convolutional architectures in vision tasks, which is also present  for their group equivariant counterparts. In other words, our networks do not build upon these architectures.
We note that this performance gap is not unique to our work but universal to self-attention. Self-attention still has problems in outperforming convolutional nets for visual data. In fact this is a very active research direction, for which alternatives are also submitted to this ICLR 2021 (with good assessment scores) https://openreview.net/forum?id=YicbFdNTTy.  However, shall self-attention networks outperform convolutional networks, we expect that by making them group equivariant additional improvements can be obtained.
* We included results from attentive G-CNNs (Romero et. al. 2020a) in our tables (**R1**).
* We fixed a typo in the results of the **Z2_CNN** reported in the CIFAR-10 Table. The results are 90.56%, 1.37M instead of 91.08%, 1.44M.

**Appendix**
* We included a new section in Appx. E, **Current empirical aspects of scale equivariant self-attention**, where we briefly discuss current practical aspects on the implementability of GSA-Nets for the scaling group. (**R4**)

**Important**: If we missed something you find important in our update please let us know. We will do our best to include it in our work.

We will now address each of your individual concerns and questions as responses to your reviews.

Best regards,
The authors.

---

> ### Author Response · Authors · 2020-11-24
> **Thank you for the reviews**
>
> With this discussion period coming to an end, we would like to thank the reviewers for the time they invested in reviewing our paper and the insightful suggestions they made to improve our work.
>
> Best regards,
>
> The authors.

---

### Comment · ~Amelie_Schreiber1 · 2023-05-10
**Equivariance of Group Self-Attention Proof Issue**

It is stated in the final proof of the paper that the **equivariance** of the group self-attention layer is due to the positional encoding being **invariant** to the group action. In particular it is stated that $L_g[\rho]((i, h_1), (j, h_2)) = \rho((i, h_1), (j, h_2))$. I do not believe this is true due to the following computation (but I would very much like to be wrong, as this paper is beautiful):



$L_g[\rho]((i, h_1), (j, h_2)) = L_yL_h[\rho]((i, h_1), (j, h_2))$

$\quad \quad \quad \quad \quad \quad =L_h [ L_y [\rho]] ((i, h_1), (j, h_2))$

$\quad \quad \quad \quad \quad \quad =L_h[ \rho] ((x^{-1}(x(i)-y), h_1), (x^{-1}(x(j)-y), h_2))$

$\quad \quad \quad \quad \quad \quad =\rho((x^{-1}(h^{-1}(x(i)-y)), h^{-1}h_1), (x^{-1}(h^{-1}(x(j)-y)), h^{-1}h_2))$

$\quad \quad \quad \quad \quad \quad =\rho^P(h^{-1}(x(j)-y) - h^{-1}(x(i)-y) , (h^{-1}h_1)^{-1}(h^{-1}h_2))$

$\quad \quad \quad \quad \quad \quad =\rho^P(h^{-1}(x(j)-x(i)) , (h^{-1}h_1)^{-1}(h^{-1}h_2))$

$\quad \quad \quad \quad \quad \quad =\rho^P(h^{-1}(x(j)-x(i)), h_1^{-1}hh^{-1}h_2)$

$\quad \quad \quad \quad \quad \quad =\rho^P(h^{-1}(x(j)-x(i)), h_1^{-1}h_2) \quad \text{notice the lack of a factor of } h^{-1} \text{ in front of } h_1^{-1}h_2 \text{ here}$

$\quad \quad \quad \quad \quad \quad \neq \rho((i, h_1), (j, h_2)) \quad \text{unless } h^{-1}(x(j)-x(i)) = x(j)-x(i)$



I have carefully checked the proofs of permutation equivariance (with no positional encodings), and the proof of translation equivariance of attention (with relative positional encodings) for the usual self-attention mechanism (*not* lifting or group self-attention). So for these I feel **very** confident in saying they are **flawless** proofs. The issue in the above calculation is only for the final two proofs in the paper (Proof of Claim 5.1 and 5.2) and their claim that the positional encoding is **invariant** to the action of $G$, that is, the claim that the equivariance holds due to the claim that $L_g[\rho]((i, h_1), (j, h_2)) = \rho((i, h_1), (j, h_2))$. This effects the proofs of Claim 5.1 (lifting self-attention) and the proof of Claim 5.2 (group self-attention). Using the above calculations, we see this is not true in general. In particular, we have $L_g [\rho] ((i, h_1), (j, h_2)) = \rho^P(h^{-1}(x(j)-x(i)), h_1^{-1}h_2) \neq \rho((i, h_1), (j, h_2))$ unless $h^{-1}(x(j)-x(i)) = x(j)-x(i)$. If $h^{-1}(x(j)-x(i)) = x(j)-x(i)$, then the claim of **invariance** of the positional encoding holds.

---

> ### Comment · ~Kaifan_Yang1 · 2023-11-17
> **We also find this and addressed it**
>
> We also identified the flaw and addressed it by proposing a new positional encoding as
> $$\rho((i,\tilde{\mathcal{h}}), (j,\hat{\mathcal{h}})) = \rho^{P}(x(j)-x(i),\tilde{\mathcal{h}}\hat{\mathcal{h}}^{-1}\tilde{\mathcal{h}})$$. The proof can be found in our UAI 2023 paper "$E(2)$-Equivariant Vision Transformer". We have also performed extensive experiments to demonstrate our theories. The code can be found at https://github.com/ZJUCDSYangKaifan/GEVit.

---

### Decision · Program_Chairs · 2021-01-07
**Final Decision**

**Decision:**

Accept (Poster)

**Comment:**

The paper introduces group equivariant self attention networks constructed by defining positional encoding that are invariant to the group action considered. This is related to equivariance in set networks . The work is sound and the idea of infusing the inductive bias via the positional encoding is  interesting and leads to improvement  results when comparing transformers with equivariance and without it, nevertheless more work needs to be done to bridge the gap in terms of performance with CNNs as pointed by the reviewers.  Authors made an admirable efforts in the rebuttal and in the revision of the paper clarifying most of the reviewers questions and concerns.   Accept